

# Atmospheric photo-oxidation of myrcene: OH reaction rate constant, gas phase oxidation products and radical budgets

Zhaofeng Tan[1], Luisa Hantschke[1], Martin Kaminski[1,a], Ismail-Hakki Acir[1,b], Birger Bohn[1], Changmin Cho[1], Hans-Peter Dorn[1], Xin Li[1,c], Anna Novelli[1], Sascha Nehr[1,d], Franz Rohrer[1], Ralf Tillmann[1], Robert Wegener[1], Andreas Hofzumahaus[1], Astrid Kiendler-Scharr[1], Andreas Wahner[1], and Hendrik Fuchs[1]

[1]Institute of Energy and Climate Research, IEK-8: Troposphere, Forschungszentrum Jülich GmbH, Jülich, Germany
[a]now at: Federal Office of Consumer Protection and Food Safety, Department 5: Method Standardisation, Reference Laboratories, Resistance to Antibiotics, Berlin, Germany
[b]now at: Institute of Nutrition and Food Sciences, Food Science, University of Bonn, Bonn, Germany
[c]now at: State Key Joint Laboratory of Environmental Simulation and Pollution Control, College of Environmental Sciences and Engineering, Peking University, Beijing, China
[d]now at: European University of Applied Sciences, Brühl, Germany

Correspondence: Hendrik Fuchs (h.fuchs@fz-juelich.de)

**Abstract.** The photo-oxidation of myrcene, a monoterpene species emitted by plants, was investigated at atmospheric conditions in the outdoor simulation chamber SAPHIR. The chemical structure of myrcene consists of one moiety that is a conjugated $\pi$-system (similar to isoprene) and another moiety that is a triple-substituted olefinic unit (similar to 2-methyl-2-butene). Hydrogen shift reactions of organic peroxy radicals ($RO_2$) formed in the reaction of isoprene with atmospheric OH radicals are

known to be of importance for the regeneration of OH. Structure-activity relationships (SAR) suggest that similar hydrogen shift reactions like in isoprene may apply to the isoprenyl part of $RO_2$ radicals formed during the OH oxidation of myrcene. In addition, SAR predicts further isomerization reactions that would be competitive with bi-molecular $RO_2$ reactions for chemical conditions that are typical for forested environments with low concentrations of nitric oxide. Assuming that OH peroxy radicals can rapidly interconvert by addition and elimination of $O_2$ like in isoprene, bulk isomerization rate constants of $0.21\,s^{-1}$

and $0.097\,s^{-1}$ ($T = 298\,K$) for the 3 isomers resulting from the 3'-OH and 1-OH addition, respectively, can be derived from SAR. Measurements of radicals and trace gases in the experiments allowed to calculate radical production and destruction rates, which are expected to be balanced. Largest discrepancies between production and destruction rates were found for $RO_2$. Additional loss of organic peroxy radicals due to isomerization reactions could explain the observed discrepancies. The uncertainty of the total radical ($RO_x$=OH+$HO_2$+$RO_2$) production rates were high due to the uncertainty in the yield of radicals

from myrcene ozonolysis. However, results indicate that radical production can only be balanced, if the reaction rate constant of the reaction between hydroperoxy ($HO_2$) and $RO_2$ radicals derived from myrcene is lower (0.9 to $1.6 \times 10^{-11}\,cm^3 s^{-1}$) than predicted by SAR. Another explanation of the discrepancies would be that a significant fraction of products (yield: 0.3 to 0.6) from these reactions include OH and $HO_2$ radicals instead of radical terminating organic peroxides. Experiments also allowed to determine the yields of organic oxidation products acetone (yield: $0.45 \pm 0.08$) and formaldehyde (yield: $0.35 \pm 0.08$). Ace-

tone and formaldehyde are produced from different oxidation pathways, so that yields of these compounds reflect the branching ratios of the initial OH addition to myrcene. Yields determined in the experiments are consistent with branching ratios expected





from SAR. The yield of organic nitrate was determined from the gas-phase budget analysis of reactive oxidized nitrogen in the chamber giving a value of $0.13 \pm 0.03$. In addition, the reaction rate constant for myrcene + OH was determined from the measured myrcene concentration yielding a value of $(2.3 \pm 0.3) \times 10^{-10}\ \mathrm{cm^3 s^{-1}}$.

# 1   Introduction

Monoterpenes are emitted from vegetation accounting for approximately $160\,\mathrm{Tg}$ of the $1000\,\mathrm{Tg}$ of biogenic volatile organic compounds (VOCs) that are released into the atmosphere per year (Guenther et al., 2012; Sindelarova et al., 2014). Monoterpenes are highly reactive to the major oxidants in the atmosphere, hydroxyl radicals (OH), ozone ($O_3$) and nitrate radicals ($NO_3$) (Atkinson and Arey, 2003) and thus play an important role in ozone and secondary organic aerosol formation (Xu et al., 2015; Zhang et al., 2018; Schwantes et al., 2020).

Myrcene emissions contribute to the total biogenic monoterpene emissions in the range of 2 to 10 % (Guenther et al., 2012). Emission rates highly depend on the type of tree and season of the year (Helmig et al., 2013). In addition, there are hints for anthropogenic sources from the analysis of the composition of indoor air (Kostiainen, 1995). Few studies have been conducted to investigate the oxidation of myrcene (Deng et al., 2018; Atkinson, 1997; Reissell et al., 2002; Kim et al., 2011). In these studies, acetone, formaldehyde and $4-\mathrm{vinyl-pentenal}$ have been identified as major oxidation products from the reaction of myrcene with OH, but yields determined in these studies vary. Lee et al. (2006) reported an organic nitrate yield of 10 % from the oxidation of myrcene by direct measurements using mass spectrometry. The reaction rate constant of myrcene + OH was determined to be $(2.1 \pm 0.2) \times 10^{-10}$ (Atkinson et al., 1986) and $(3.4^{+1.5}_{-1.0}) \times 10^{-10}$ (Hites and Turner, 2009) at room temperature with a discrepancy of up to 60 %. These results demonstrate the photo-oxidation of myrcene requires further investigation.

There is no detailed chemical mechanism of myrcene degradation by OH in literature. The acyclic structure of myrcene consists of two parts, an isoprenyl part ($CH_2{=}CH{-}C({=}CH_2)CH_2{-}$ moiety) and another part that is structurally similar to $2-\mathrm{methyl-2-butene}$ ($(CH_3)_2)C{=}CH{-}CH_2{-}$ moiety) (Fig. 1). Recent investigations of the oxidation of isoprene (Fuchs et al., 2013; Peeters et al., 2014; Wennberg et al., 2018; Novelli et al., 2020) revealed that organic peroxy radicals ($RO_2$) formed after the attack of OH to isoprene can rapidly interconvert, so that fast H-atom migration reactions of specific $RO_2$ isomers with initially low yield can significantly gain in importance. These reactions impact atmospheric chemistry most, if they become competitive with bi-molecular loss reactions such as reactions with nitric oxide (NO) and hydroperoxy radicals ($HO_2$). In the case of isoprene, these isomerization reactions can eventually regenerate OH radicals, so that a high OH regeneration efficiency of 0.5 can be sustained in the atmosphere also in regions, where radical termination reactions have been thought to dominate the fate of radicals (Novelli et al., 2020). Therefore, radical regeneration from isomerization reactions helped partly to explain observations of unexpectedly high OH concentrations in field experiments (Lelieveld et al., 2008; Hofzumahaus et al., 2009; Whalley et al., 2011). Radical regeneration from $RO_2$ isomerization reactions have also been shown to play a role in the oxidation of methacrolein (Fuchs et al., 2014; Crounse et al., 2012), 3-pentanone (Crounse et al., 2013), glyoxal (Lockhart et al., 2013), n-hexane, 2-hexanol (Praske et al., 2018), hydroxymethyl hydroperoxides (Allen et al., 2018), and 2-hydroperoxy-2-methylpentane (Praske et al., 2019).





In this study, the oxidation of myrcene by OH was investigated in the atmospheric simulation chamber SAPHIR (Simulation of Atmospheric Photochemistry In a Large Reaction Chamber) at Forschungszentrum Jülich. Experiments were performed under controlled conditions with concentrations of trace gases and radicals typical for atmospheric conditions. Experiments aim for adding to the determination of a degradation mechanism for myrcene.

## 2   Oxidation mechanism of myrcene

Myrcene is an acyclic hydrocarbon with three unsaturated carbon double bonds to which OH preferentially adds. One end of the myrcene molecule is structurally similar to isoprene and the other end is similar to 2-methyl-2-butene (Fig. 1). The myrcene oxidation by OH forms ten isomers of peroxy radicals ($MyO_2$) (Fig. 2). Structure-activity relationships (SAR) by Peeters et al. (2007) suggest that OH preferably adds to the double bound of the isolated $-CH=C(CH_3)_2$ moiety producing two peroxy radical species with yields of 17 % (7-OH-6-OO radical) and 31 % (6-OH-7-OO radical) (Fig. 2). When OH adds

to the conjugated carbon double bonds, six isomeric hydroxy peroxy radicals are formed like in the OH or $NO_3$ oxidation of isoprene (Peeters et al., 2009, 2014; Vereecken et al., 2021a). Addition to the $C_{3'}$ position yields two allyl-like hydroxy radical isomers producing E-3'-OH-1-OO, 3'-OH-3-OO and Z-3'-OH-1-OO peroxy radicals after the subsequent addition of $O_2$. A total yield of 31 % is estimated by SAR (Peeters et al., 2007) for these three isomers. Similarly, OH addition to the $C_1$ position followed by addition of $O_2$ to the allyl radical structure leads to the formation of E-1-OH-3'-OO, 1-OH-2-OO

and Z-1-OH-3'-OO radicals with a total yield of 17 %. In the isoprene mechanism developed by Peeters et al. (2009, 2014); Novelli et al. (2020); Vereecken et al. (2021a), a central element is the fast interconversion reactions between the OH-adducts and corresponding OH peroxy radicals that proceed by addition and elimination of $O_2$. Similar reactions are expected for the isoprene structure in myrcene (Fig. 2) establishing a coupled equilibrium between $MyO_2$ isomers.

For OH attack on the $C_2$ or $C_3$ position of myrcene, the resulting OH adducts are not resonance stabilized and are hence

not as favorable as the allylic-type radicals produced from $C_1$ or $C_{3'}$ addition. Yields of these $MyO_2$ radicals are expected to be less than 2 % and therefore the chemistry of these two minor isomers is not further discussed in this work.

All $MyO_2$ isomers can undergo bi-molecular reactions with NO, $HO_2$ and other $RO_2$ species. Specific reaction rate constants have not been measured, but values are expected to be within the range of typical rate constants for organic peroxy radicals. For example, SAR by Jenkin et al. (2019) suggests a value of $2.2 \times 10^{-11}\,cm^3s^{-1}$ ($T = 298\,K$) of the reaction rate

constants for the reaction of monoterpene derived $RO_2$ (including $MyO_2$) with $HO_2$. Reaction products of the reaction of $RO_2$ with $HO_2$ are expected to be organic peroxides.

The reaction of the 6-OH-7-OO and 7-OH-6-OO radicals with NO produce mainly 4-vinyl-4-pentenal, acetone and $HO_2$ (Fig. 3) as shown in epxerimental studies (Orlando et al., 2000; Reissell et al., 2002; Lee et al., 2006) as well as predicted by SAR (Vereecken and Peters, 2009, 2010). Reissell et al. (2002) and Lee et al. (2006) suggested additional pathways for

the alkoxy radical that is formed in the reaction with NO of these two $MyO_2$ species. These pathways would include rearrangement reactions followed by reaction with NO and fragmentation. This was suggested to explain the observation of organic species with various masses (m/z=71, m/z=113, m/z=115) detected by the PTR-MS instrument in Lee et al. (2006).





Similarly, Böge et al. (2013) suggested a reaction pathway of 6-OH-7-OO and 7-OH-6-OO radicals to explain observed ter-penylic acid in their experiments. The reaction of the other two most abundant radicals, 1-OH-2-OO and 3'-OH-3-OO, with NO would lead to the formation of $HO_2$ and formaldehyde together with 2-methyldiene-6-methyl-5-heptenal and 1-vinyl-5-methyl-4-hexanone, respectively (Fig. 3).

In addition to bi-molecular reactions, uni-molecular can be of importance for specific $MyO_2$. Due to the similarity of the conjugated double bond structure in myrcene and isoprene, it can be expected that H-shift reactions found to be important in isoprene (Peeters et al., 2009, 2014; Fuchs et al., 2013; Wennberg et al., 2018; Novelli et al., 2020) apply for corresponding $MyO_2$ radicals. Therefore, Z-3'-OH-1-OO and Z-1-OH-3'-OO radicals are expected to undergo $\alpha$-OH 1,6-H migration followed by $O_2$ addition leading to hydroperoxy peroxy radicals (Fig. 4). As a first approximation, reaction rate constants can be assumed to be in the order of $1\,s^{-1}$ at 298 K like for corresponding reactions in isoprene (Peeters et al., 2014). Similar products from further isomerization and decomposition reactions could be expected as predicted for isoprene (Peeters et al., 2014). This would also lead to the regeneration of HOx radicals.

According to SAR by Vereecken and Nozière (2020), E-3'-OH-1-OO undergo an allylic 1,7 H-shift (Fig. 4). The reaction rate constant is estimated by this SAR to be high with a value of $8\,s^{-1}$ ($T = 298\,K$). In addition, E-1-OH-3'-OO can undergo an allylic 1,6 H-shift isomerization reaction with a fast isomerization reaction rate constant of $2\,s^{-1}$ ($T = 298\,K$). Products likely undergo fast ring-closure reactions on the dimethyl double bond with rates on the order of $1\,s^{-1}$ (pers. comm. Vereecken , 2021b).

As a consequence of the equilibration between different $MyO_2$ isomers that originate from the 3'-OH or 1-OH addition, a significant fraction of the $MyO_2$ can be removed through the H-shift reaction channels, if rates of competing bi-molecular reactions of all $RO_2$ isomers are low enough. In order to estimate effective bulk $MyO_2$ isomerization reaction rates, the distribution of $MyO_2$ isomers is estimated by using reaction rate constants for the oxygen addition and elimination reactions recommended for isoprene (Novelli et al., 2020). This results in a total bulk $MyO_2$ loss rate of approximately $0.21\,s^{-1}$ and $0.097\,s^{-1}$ ($T = 298\,K$) for the 3 isomers resulting from the 3'-OH and 1-OH addition, respectively. This means that isomerization reactions are becoming competitive for nearly half of the total $MyO_2$ (Fig. 1) isomers for chemical conditions with NO mixing ratios lower than 1 ppbv. It is worth noting that isomerization reactions rate constants have a strong temperature dependence, so that their impact can be significantly enhanced at higher temperatures. However, all rate constants by SAR predictions have also a high uncertainty of at least a factor of two and the uncertainty might be as high as a factor of 10 (Vereecken and Nozière, 2020).

## 3 Methods

### 3.1 Experiments in the SAPHIR chamber

The experiments were conducted in the outdoor atmospheric simulation chamber SAPHIR. SAPHIR has a cylindrical shape with double walls made of Teflon (FEP) film (length: 18 m diameter: 5 m, volume: $270\,m^3$). The space between the double walls is permanently purged with clean air to avoid diffusion of impurities from ambient air into the inner chamber. The walls



are transmissive for the entire solar UV and visible spectrum. The chamber is operated with synthetic air that is produced from evaporated liquid nitrogen and oxygen of highest purity (Linde, purity> 99.99990 %). It is kept at a slight overpressure of 35 Pa that is maintained by a replenishment flow, which compensates for leakages and the sampling flow of analytical instruments. As a consequence, trace gases are diluted with a rate constant that is equivalent to a lifetime of approximately

17 hours. The air in the chamber can be exposed to sunlight by opening a shutter system. When the chamber film is exposed to solar radiation, nitrous acid (HONO), formaldehyde and acetone are released. The source strengths range between 100 to 200 pptv/h. The photolysis of HONO leads to a continuous increase of nitrogen oxide concentrations in the chamber and is a significant source for hydroxyl radicals (OH). More details of the SAPHIR can be found in previous publications (Bohn et al., 2005; Rohrer et al., 2005, e.g.).

In total 4 experiments investigating the oxidation of myrcene by OH were performed (Table 1), two of which were done at medium levels of nitric oxide (NO) (18 August 2012: 0.18 to 0.43 ppbv NO (Fig. 5) and 22 August 2012: 0.15 to 0.30 ppbv of NO (Fig. S1 Supplement)) while lower NO mixing ratios below 0.11 ppbv were achieved in the other two experiments (17 July 2013, Fig. S2 Supplement, and 18 July 2013, Fig. S3 Supplement).

The experimental procedure was similar in all experiments. The chamber was cleaned in the night before the experiment by
flushing the chamber with a high flow of synthetic air to remove all trace gases from previous experiments. Figure 5 shows, as an example, time series of trace gases for the experiment conducted on 22 August 2012. Experiments started in the morning with humidification of the chamber air in the dark. This was achieved by flushing evaporated Milli-Q water into the chamber together with a high flow of synthetic air. The chamber air was exposed to sunlight approximately for 1 hour without the presence of any additional reactant to determine the source strengths of chamber sources for formaldehyde for the specific
experiment. In the experiments in 2013, approximately 50 ppbv ozone produced by a silent discharge ozonizer (O3onia) was injected to suppress NO in the reaction with $O_3$.

Air mixtures of myrcene (Sigma-Aldrich, purity 99 %) were premixed in a canister (stainless steel with Silconert coating). The mixture was injected into the chamber air by calibrated mass flow-controllers to reach myrcene mixing ratios of several ppbv. Injections were done 2 times (3 ppbv each) in experiments with medium NO and four injections with smaller concen-
trations (approximately 2 ppbv) were done in the other experiments.

Additional reference experiments were performed using the same procedure like for the experiments with myrcene, but either with one injection of 150 ppmv of methane (29 May 2020 (Fig. S4 Supplement) and 10 July 2013) or three injections of 5 ppbv of $\alpha$-pinene (03 September 2019 (Fig. S5 Supplement)). These experiments were used to evaluate the accuracies of the procedures that were used to analyse the organic nitrate formation and radical budgets in the experiments with myrcene.

**3.2 Measurement of trace gas concentrations**

Table 2 summarizes properties of instruments used in this work. The set of instruments was similar like used in previous experiments investigating the photochemistry of organic compounds (Kaminski et al., 2017; Fuchs et al., 2018; Novelli et al., 2018; Rolletter et al., 2019). Therefore, only a brief description is given here. Ozone was detected by a UV photometer (Ansyco). NO concentrations were measured by a chemiluminescence instrument (Eco Physics) that also detected nitrogen





dioxide ($NO_2$) after conversion to NO in a photolytic converter. Methane, water vapour and carbon monoxide concentrations were measured by a cavity ring-down instrument (Picarro). Nitrous acid (HONO) was detected by a custom-built long-path absorption photometry (LOPAP) (Li et al., 2014). Photolysis frequencies were derived from solar actinic flux measurements by a spectroradiometer outside the chamber. Calculations take into account the reduction of radiation by the chamber construction elements and the Teflon film (Bohn and Zilken, 2005).

Myrcene was detected by gas chromatography coupled with a flame ionization detector (GC-FID) (Wegener et al., 2007) and by a proton-transfer-reaction mass-spectrometer (PTR-MS, Ionicon). The PTR-MS instrument was not calibrated for myrcene. Therefore, the signal was scaled to match the GC-FID measurements and used because of its higher time resolution (40 s) compared to that of GC-FID (30 min). Acetone and acetaldehyde were also measured by GC-FID. Formaldehyde (HCHO) measurements were performed by a Hantzsch instrument (AeroLaser) or by differential optical absorption spectrometry (DOAS).

OH reactivity ($k_{OH}$), which is the pseudo first-order rate constant of the OH radical loss, was measured by a laser flash photolysis - laser induced fluorescence instrument (Lou et al., 2010; Fuchs et al., 2017).

### 3.3    Measurement of radical concentrations

Measurements of OH radicals were performed by differential optical absorption spectrometry (DOAS) (Dorn et al., 1995) and laser-induced fluorescence (LIF) (Tan et al. (2017) and references therein). The DOAS instrument was not available in the

experiment on 16 August 2012. In the experiment 6 days later (22 August 2012), OH concentrations measured by the LIF instrument were consistently 20 % higher than those measured by DOAS that is considered to be an absolute standard. The difference is larger than the combined $1\sigma$ uncertainty (14.5 %) of the measurements and is probably caused by a calibration error of the LIF instrument. This assumption is supported by the observed decay of myrcene which is caused by the reaction with OH. The decay can be accurately described with LIF data if these are reduced by 20 % in both experiments (Fig. S6

Supplement, Section 6.1). Therefore, OH measurements by LIF were scaled by a factor of 0.8 for both experiments. For the experiments in 2013, OH measurements by LIF and DOAS well agreed within 5 %. For the analysis of this work, OH measurements by the DOAS instrument are used if available.

In addition to OH, peroxy radicals ($HO_2$ and $RO_2$) can be detected by the LIF instrument after chemical conversion to OH. The conversion of $HO_2$ radicals is accomplished by adding excess NO in a second low-pressure fluorescence cell, in which the

sum of $HO_2$ and OH concentrations is measured (Fuchs et al., 2011). The measurement of $RO_2$ is accomplished in a 2-stage system (ROxLIF) in which all atmospheric $RO_x$ radicals are first converted to $HO_2$ by added NO and CO in a flow reactor (Fuchs et al., 2008). This is followed by $HO_2$ to OH conversion with additionally added NO in a fluorescence cell. In this case, $RO_2$ is determined as the difference of $RO_x$ and OH + $HO_2$. The operational parameters of the reactor are optimized to maximize the detection sensitivity for methyl peroxy radicals ($CH_3O_2$). However, it was found in previous studies that

specific $RO_2$ species from other VOCs may be converted less efficiently in the reactor, if their conversion to $HO_2$ needs more reaction steps and therefore more reaction time than in case of $CH_3O_2$ (Fuchs et al., 2008). Alternative explanation for the lower sensitivity could be caused by the reversibility of the $MyO_2$ formation at low $O_2$ in the reactor, which slow down the conversion to $HO_2$.



For the evaluation of data in this work, the ROxLIF instrument was calibrated for $CH_3O_2$ and myrcene $RO_2$ radicals ($MyO_2$) that were produced by reaction of OH with the corresponding VOC in the radical calibration source as described by Fuchs et al. (2008). The detection sensitivity was only half as high for $MyO_2$ compared to $CH_3O_2$. The lower sensitivity could hint that some specific $MyO_2$ isomers are not efficiently converted to $HO_2$ for conditions inside the conversion reactor. The difference in the detection sensitivity for $MyO_2$ and other $RO_2$ species like $CH_3O_2$ adds to the uncertainty of measurements, because a mixture of different peroxy radicals is present during the experiments and the exact distribution of $RO_2$ species is not known. For the analysis in this work, the fraction of $MyO_2$ is estimated as described in the next subsection, in order to account for the lower sensitivity.

### 3.4 VOC reactivity and $RO_2$ speciation in myrcene experiments

The total OH reactivity ($k_{OH}$) that was measured in the experiments consists of reactivity from organic and inorganic compounds. The reaction of OH with most of the organic compounds leads to the formation of $RO_2$ radicals. Therefore, it is useful to distinguish between OH reactivity from those compounds unmeasured ($k_{OHVOC}$) and OH reactivity calculated from measured concentrations of inorganic compounds (CO, NO, $NO_2$) and formaldehyde:

$$k_{OHVOC} = k_{OH} - (k_8[HCHO] + k_9[CO] + k_{12}[NO_2] + k_{13}[NO]) \tag{1}$$

$k_i$ are bimolecular rate constants of reactions $R_i$ listed in Table 3. For the reference experiment with methane, only methane contributed to the VOC reactivity, because its oxidation product is formaldehyde, which does not produce $RO_2$ in the reaction with OH. Therefore, methyl peroxy radicals are the only $RO_2$ radical expected in that experiment. For the experiments with myrcene, the VOC reactivity includes the reactivity from myrcene and from partly unmeasured oxygenated organic compounds (OVOC) that are products of the myrcene oxidation. The OH reactivity from myrcene ($k_{OHmyrcene}$) can be calculated from measured myrcene concentrations and the rate constant of its reaction with OH ($k_7$, Table 3):

$$k_{OHmyrcene} = k_7[myrcene] \tag{2}$$

Assuming that each OH reaction with an organic compound except formaldehyde in the experiments in this work leads to the formation of one $RO_2$ radical and all $RO_2$ radicals have similar chemical lifetimes, the distribution of $RO_2$ species from different VOCs is similar to the distribution of the OH reactivity from the VOCs. Therefore, the concentration of $RO_2$ derived from myrcene ($MyO_2$) can be approximated by scaling the total measured organic peroxy radical concentration ($[RO_2]_m$) that is determined assuming the same instrument sensitivity for all $RO_2$ with the ratio of OH reactivity from myrcene ($k_{OHmyrcene}$) to the total OH reactivity from VOCs ($k_{OHVOC}$). Taking also into account that the sensitivity of the instrument for $MyO_2$ ($S_{MyO_2}$) is reduced compared to the sensitivity for $CH_3O_2$ ($S_{CH3O2}$), the concentration of $MyO_2$ can be calculated from observed quantities:



$$[\mathrm{MyO_2}] = \frac{1}{\frac{S_{\mathrm{MyO2}}}{S_{\mathrm{CH3O2}}} + \frac{k_{\mathrm{OHVOC}}}{k_{\mathrm{OHmyrcene}}} - 1}[\mathrm{RO_2}]_m \tag{3}$$

$\mathrm{RO_2}$ species originating from other hydrocarbons than myrcene ($\mathrm{RO_{2,other}}$) can be calculated using the remaining fraction
of the VOC reactivity:

$$[\mathrm{RO_{2,other}}] = [\mathrm{MyO_2}]\left(\frac{k_{\mathrm{OHVOC}}}{k_{\mathrm{OHmyrcene}}} - 1\right) \tag{4}$$

Equations 3 and 4 allow calculating a more realistic total $\mathrm{RO_2}$ concentration by taking the sum of $\mathrm{MyO_2}$ and $\mathrm{RO_{2,other}}$ instead of using the measured $\mathrm{RO_2}$ concentration, if the same instrument sensitivity for all $\mathrm{RO_2}$ is assumed. An example of the result of this calculation is shown in Fig. 5 demonstrating that $\mathrm{MyO_2}$ was the dominant $\mathrm{RO_2}$ species right after each myrcene
injection. However, these values have a high uncertainty that cannot easily be quantified, because, for example, the detection sensitivity for $\mathrm{RO_2}$ produced in the reaction of OH with oxidation products may also differ from that of methyl peroxy radicals. The uncertainty of calculated $\mathrm{RO_2}$ concentrations is lowest after each injection of myrcene, because the total OH reactivity and therefore $\mathrm{RO_2}$ production is dominated by myrcene. Therefore, the analysis of radical production and destruction rates in this work (Section 6.4) focuses on the times right after each myrcene injection.

## 4 Experimental determination of organic nitrate yields

The determination of organic nitrate yields from photochemical VOC oxidation makes use of a budget analysis of the sum of $\mathrm{NO_x}$ and HONO in the chamber air. In the following, the sum of NO, $\mathrm{NO_2}$ and HONO concentrations is defined as $\mathrm{NO_y^*}$, which can be calculated from measurements of these species (Table 2).

In the sunlit SAPHIR chamber, the only source of reactive nitrogen is the emission of HONO from the chamber film
(Section 3.1). The source strength $Q(\mathrm{HONO})$ is variable and depends on solar ultraviolet radiation, temperature and relative humidity (Rohrer et al., 2005). HONO is photolyzed to OH and NO (R1), which is further oxidized to $\mathrm{NO_2}$ by reactions with $\mathrm{O_3}$, $\mathrm{HO_2}$ (Reaction R5), and $\mathrm{RO_2}$ (Reaction R11). The sum of of $\mathrm{NO_y^*}$ is chemically lost in the chamber by reactions forming nitric acid ($\mathrm{HNO_3}$) (Reaction R12) and organic nitrates ($\mathrm{RONO_2}$) (R14). In addition, the $\mathrm{NO_y^*}$ species are removed ($L(\mathrm{NO_y^*})$) by transport due to the replenishment flow that compensates for chamber leakage and gas sampling of analytical instruments
with a first-order rate constant of $k_d \approx 1.6 \times 10^{-5}\,\mathrm{s^{-1}}$ (Section 3.1):

$$L(\mathrm{NO_y^*}) = ([\mathrm{NO}] + [\mathrm{NO_2}] + [\mathrm{HONO}])\,k_d \tag{5}$$

The dilution rate is monitored by the input flow rate of synthetic air which yield an accuracy of $k_d$ better than $1\,\%$. Hence, the uncertainty due to the dilution is neglected due to its small contribution. The concentration of $\mathrm{NO_y^*}$ at a given time is then determined by the difference between the time-integrated production and loss terms.





$$[NO_y^*] = \int_t \left( Q(HONO) - k_{12}[OH][NO_2] - k_{14}[RO_2][NO] - L(NO_y^*) \right) dt \tag{6}$$

It is assumed that the formation of $HNO_3$ and $RONO_2$ are effective sinks for $NO_x$ and that reformation of $NO_x$ by their reactions with OH or photolysis does not play a role. This assumption is justified because the low reaction rate constant with OH and small absorption cross sections of $HNO_3$ and $RONO_2$ (Burkholder et al., 2020; Browne et al., 2014) lead to their lifetimes in the range of several days, much slower than the time scale of the experiments. The loss of $NO_2$ due to reaction with

ozone which forms nitrate radicals ($NO_3$) is neglected in Equation 6, because the $NO_3$ radical is efficiently converted back to $NO_2$ by photolysis and reaction with NO in the photo-oxidation experiments in this work. The formation of other oxidized nitrogen species such as peroxy nitric acid ($HNO_4$) and acetyl peroxy nitrate (PAN) is also assumed to be negligible because their mixing ratios are expected to be only a few tens $pptv$ for conditions of the experiments.

The formation rate of HONO that is within the range of a few hundred $pptv/h$ in the chamber experiments can be determined

from measurements of OH, NO, HONO and j(HONO) by assuming a photo-stationary state for the HONO concentration:

$$\frac{d[HONO]}{dt} = Q(HONO) - j_{HONO}[HONO] + k_{OH+NO}[OH][NO] = 0$$
$$Q(HONO) = j_{HONO}[HONO] - k_{13}[OH][NO] \tag{7}$$

For the experimental conditions ($j(HONO) = 8 \times 10^{-4}\,s^{-1}$) photo-stationary state is reached within approximately 20 minutes. On this time scale, the dilution of HONO by the chamber replenishment flow is only 2 % and is therefore neglected.

The uncertainty of the HONO formation rate (Equation 7 is dominated by the uncertainty in the HONO measurement (22 %, Table 3).

Using measured concentrations of NO, $NO_2$ and HONO, the concentration of NO that is converted to organic nitrates during the experiment ($\Delta RONO_2$) is determined by the balance of nitrogen oxide concentrations:

$$\Delta[RONO_2] = \int_t k_{14}[RO_2][NO]dt$$

$$= \Phi_{RONO2} \int_t (k_{11} + k_{14})[RO_2][NO]dt$$

$$= \int_t \left( Q(HONO) - k_{12}[OH][NO_2] - L(NO_y^*) \right) dt - ([NO] + [NO_2] + [HONO]) \tag{8}$$

The reaction yield of organic nitrates ($\Phi_{RONO2}$) can be then derived as slope of a linear fit (Equation 8). The accuracy of the yield is mainly determined by the accuracy the kinetic reaction rate constants and measurements that vary between the specific experiments.





## 4.1 Test experiments with methane and $\alpha$-pinene

In order to test the method described above, organic nitrate yields from the photo-oxidation of methane and $\alpha$-pinene were experimentally determined and compared with literature values. NO mixing ratios were between 100 to 300 pptv in both experiments, so that at least 60 % of the $RO_2$ radicals reacted with NO.

Figure 6 shows time series of $NO_y$ species in $\alpha$-pinene and methane experiments. In the methane experiment, the total mixing ratio of $NO_y$ species calculated from the HONO chamber source (Equation 7) is around 1.3 ppbv and is mainly explained by measured $NO_2$ and NO concentrations. In the experiment with $\alpha$-pinene, the calculated $NO_y$ mixing ratio increased from 1 to 1.3 ppbv within two hours. Approximately 1 ppbv was present as $NO_2$ and NO and the remaining fraction can be attributed to the formation of organic nitrates from $\alpha$-pinene. The alkyl nitrate yield is derived from Equation 8 by a linear fit as described above.

For the experiment with methane, the alkyl nitrate yield is smaller than error of the calculation ($0.00 \pm 0.04$, Fig. 6). This small value well agrees with literature values of $< 0.003$ (Scholtens et al., 1995) and $0.0039 \pm 0.0011$ (Butkovskaya et al., 2012) at room temperature.

The alkyl nitrate yield determined from the OH oxidation experiment with $\alpha$-pinene is $0.32 \pm 0.06$. The accuracy is mainly due to the uncertainty of $RO_2$ measurements. Reported literature values are 0.01 (Aschmann et al., 2002), $0.18 \pm 0.09$ (Noziére et al., 1999) and $0.26 \pm 0.07$ (Rindelaub et al., 2015). All experiments were conducted at measurement conditions where the reaction of $RO_2$ with NO was the dominant $RO_2$ reaction pathway, so that the obtained organic nitrate yields can be attributed to the yield from reaction of $\alpha$-pinene derived $RO_2$. The value determined in this work agrees well with values determined from direct measurements using FT-IR spectroscopy in experiments by Noziére et al. (1999) and Rindelaub et al. (2015) within the specified errors. In contrast, Aschmann et al. (2002) approximated the nitrate yield from mass spectrometer measurements with a high uncertainty that likely explains the much lower yield in their experiments compared to the other two studies and results in this work.

The good agreement between nitrate yields determined in the experiments with methane and $\alpha$-pinene with reported values in literature demonstrates that the applied method for the determination of alkyl nitrate formation in the chamber gives reliable results.

## 5 Experimental analysis of the chemical budgets of OH, $HO_2$, $RO_2$ and $RO_x$

The chamber experiments are also used to study the chemical budgets of OH, $HO_2$, $RO_2$, and $RO_x$ during the photochemical oxidation of myrcene. Radical production and destruction rates are determined from reaction kinetic data and measured trace gas concentrations for reactions that are known to produce or consume radicals in the experiments. Because the chemical lifetime of radicals is short (seconds to minutes), the radical concentrations are expected to be in steady state. Therefore, the total loss rates for each radical species is balanced by its total production rate. Analysing radical production and destruction rates using experimental data gives an indication if reactions taken into account can describe observations or if there are contributions from further reactions. A similar method has been used before by Tan et al. (2019, 2020) to analyse radical





budgets in atmospheric air in China. Here, the method is tested for the photochemical degradation of methane and then applied to the more complex degradation mechanism of myrcene.

## 5.1 Radical production, destruction and regeneration reactions

In the chamber experiments, OH, $HO_2$ and $RO_2$ radicals are primarily formed from photolysis of HONO (Reaction R1), $O_3$ (Reaction R2), formaldehyde (Reaction R3), and from the ozonolysis of myrcene (Reaction R4). Radical termination processes include the formation of nitrates ($HNO_3$, HONO and $RONO_2$, Reactions R12, R13 and 14) and peroxides (ROOH and $H_2O_2$, Reaction R15 and R16). The contribution from $RO_2$ self-combination reactions is neglibile and not considered here. The photolysis of peroxides that can lead to the production of radicals is neglected due to the slow photolysis frequency (typical value for example for $CH_3OOH$: $2 \times 10^{-6}\,s^{-1}$ at noontime). The rates for the primary production of $RO_x$ radicals ($P(ROx)$) and for the termination loss rate ($L(ROx)$) can be calculated as:

$$P(RO_x) = j_{HONO}[HONO] + \Phi_{OH,2}j_{O(^1D)}[O_3] + 2j_{HCHO}[HCHO] + (\Phi_{OH,4} + \Phi_{RO_2,4})k_4[VOC][O_3] \qquad (9)$$

$$L(RO_x) = (k_{12}[NO_2] + k_{13}[NO])[OH] + (k_{14}[NO] + 2k_{15}[HO_2])[RO_2] + 2k_{16}[HO_2]^2 \qquad (10)$$

For the calculation of $P(RO_x)$ and $L(RO_x)$, measured radical and trace gas concentrations are used. For the analysis of the myrcene experiments, total $RO_2$ concentrations include corrections for the reduced detection sensitivity of $MyO_2$ as explained in Section 3.4. The applied rate constants are listed in Table 3. The lumped rate constant for the reactions between $RO_2$ and $HO_2$ (R15) has a rather large uncertainty, because rate constants of different $RO_2$ species can be different by up to a factor of 4 (Jenkin et al., 2019). The uncertainty of other reaction rate constants is typically around 10 %. In addition, the accuracies of measurements (Table 2) add to the total uncertainty in the calculation of loss and production rates. In these experiments, there is a small background OH reactivity ($< 1\,s^{-1}$) which can be a permanent loss of radicals, but could also regenerate $HO_2$ and/or $RO_2$. However, this background reactivity is small compared to the OH reactivity from methane and myrcene during the experiments ($> 15\,s^{-1}$), so that it does not affect the analysis.

If production and destruction rates of single $RO_x$ species (OH, $HO_2$ and $RO_2$) are calculated, radical conversion reactions need to be additionally taken into account. The total OH loss rate ($L(OH)$) can be quantified by the product of measured OH concentrations and total OH reactivity:

$$L(OH) = [OH]k_{OH} \qquad (11)$$

The OH production rate ($P(OH)$) can be calculated from the sum of production from HONO (Reaction R1) and $O_3$ photolysis followed by water reaction (Reaction R2), ozonolysis of VOCs (Reaction R4) and radical regeneration from $HO_2$ reacting with NO (Reaction R5) or $O_3$ (Reaction R6):

$$P(OH) = j_{HONO}[HONO] + \Phi_{OH,2}j_{O(^1D)}[O_3] + \Phi_{OH,4}k_4[VOC][O_3] + (k_5[NO] + k_6[O_3])[HO_2] \qquad (12)$$





The loss of $HO_2$ ($L(HO_2)$) and $RO_2$ ($L(RO_2)$) radicals are dominated by their reactions with NO for conditions of the experiments here (Reactions R5, R11, R14) in addition to recombination of peroxy radicals including reactions of $HO_2$ with $RO_2$ (Reaction R15) and $HO_2$ self-reactions (Reaction R16):

$$L(HO_2) = (k_5[NO] + k_6[O_3] + k_{15}[RO_2] + 2k_{16}[HO_2])[HO_2] \tag{13}$$

$$L(RO_2) = ((k_{11} + k_{14})[NO] + k_{15}[HO_2])[RO_2] \tag{14}$$

The $HO_2$ production ($P(HO_2)$) rate can be calculated from the photolysis of aldehydes, of which HCHO (Reaction R3) were measured in these experiments, and reactions that convert OH or $RO_2$ to $HO_2$. OH to $HO_2$ conversion occurs in the reaction of OH with HCHO (Reaction R6), CO (Reaction R7) and $O_3$ (Reaction R8). The reaction of $RO_2$ with NO (Reaction R9) produces either $HO_2$ or organic nitrates (Reaction R14). The total $HO_2$ production rate is then calculated as:

$$P(HO_2) = 2j_{HCHO}[HCHO] + (k_6[HCHO] + k_7[CO] + k_8[O_3])[OH] + k_{11}[NO][RO_2] \tag{15}$$

$RO_2$ primary production consists of the ozonolysis of VOCs (Reaction R4). In addition $RO_2$ is produced from radical propagation reactions via the reaction of OH with VOCs. The total production rate ($P(RO_2)$) can be calculated from the VOC reactivity, $k_{OHVOC}$, (Equation 1) assuming that each reaction of a VOC with OH produces one $RO_2$ radical:

$$P(RO_2) = \Phi_{RO2} k_4[VOC][O_3] + k_{OHVOC}[OH] \tag{16}$$

### 5.2 Radical production and destruction in a test experiment with methane

The chemical radical budget analysis in SAPHIR experiments was tested in a photo-oxidation experiment with methane. The chemical oxidation mechanism of methane is much simpler than that of monoterpenes. (1) Ozonolysis reactions do not play a role. (2) Organic peroxy radicals that are formed in the chemical mechanism are methyl peroxy radicals ($CH_3O_2$), which can be accurately measured by the ROxLIF system. (3) There are recommendations for rate constants for reactions involving methyl peroxy radicals (IUPAC, 2020).

Reaction rates for the methane oxidation experiment (29 May 2020) are shown in Fig. 7. Since no DOAS OH measurements were available for this experiment, OH measurements by the LIF instrument were used (Fig. S4). Total turnover rates are similar for all single radical species (OH, $HO_2$, $RO_2$) with values between 8 and 12 ppbv/h and do not vary much over the course of the experiment. The loss of peroxy radicals is dominated by the radical regeneration reaction with NO whereas radical recombination reactions contribute less than 10 % to the entire loss of peroxy radicals. OH is nearly only lost by its reaction with methane and formaldehyde.

The production and destruction rates of total $RO_x$ are significantly smaller than those of single radical species with values that rise from 2 to 4 ppbv/h over the course of the experiment as radical regeneration reactions cancel out. The increase is due





to radicals from the photolysis of formaldehyde that is continuously produced from the chamber wall (Section 3.1) and in the reaction of OH + methane. The increase in radical production is balanced by increasing rates of peroxy radical recombination reactions and of the reaction of OH with $NO_2$. The latter is due to the increase of nitrogen oxide concentrations from the chamber source of nitrous acid.

Radical production and destruction rates of each radical species and of total $RO_x$ are roughly balanced within the uncertainty
of the calculation (Fig. 7). Maximum deviations are less than 0.5 ppbv/h for the production and destruction rates of $RO_x$ and $HO_2$. Differences are much smaller than the accuracy of the calculation (1.5 ppbv/h for $RO_x$ and 2 ppbv/h for $HO_2$). Higher deviations with values of up to 4 ppbv/h are seen for OH and $RO_2$ but opposite behaviour. The accuracy of OH and $RO_2$ radical production and destruction rates are in the range of 2 to 3 ppbv/h, which cannot explain the discrepancies. As the $RO_x$ and $HO_2$ budget are closed using measured OH concentrations, the imbalance between OH and $RO_2$ production and
destruction indicates an unknown systematic error in the conversion rate from OH to $RO_2$. One possible explanation for the observed imbalances would be that the calculated reaction rate of OH + $CH_4$ is too large. This could either be caused by the measured OH or $CH_4$ concentrations being too high, or by the applied reaction rate constant $k_{OH+CH_4}$ being too large.

The methane concentrations in the chamber were above the measurement range specified by the manufacturer (Picarro, 0-20 ppmv. However, an instrumental test of the instrument done prior to the chamber experiment showed that the stated
measurement accuracy holds at the concentrattion (150 ppmv) used in the presented experiment. A small background reacitivty of $2 \, s^{-1}$ was found before the injection of methane. The OH reactivity calculated from measured methane concentration and the reaction rate constant is about $26 \, s^{-1}$, which is consistent with the direct OH reactivity measurement considering the contribuion of background reacivity (Fig. S4). This gives the confidence on the accuracy of the methane measurement.

Another reason for the imbalances could be LIF calibration errors that are larger than given in Table 2. The specification
for OH in Table 2 was generally confirmed in previous intercomparisons with the OH-DOAS instrument. However, in some instances the LIF measurements were found to deviate more than expected. An example are the experiments on 16 Aug and 22 Aug 2012 (Fig. 5 and Fig. S1), where the LIF measurements are higher than the DOAS data by a factor 1.2 probably due to an incorrect calibration.

In the following Section, radical budgets for the chemical degradation of myrcene are investigated. In these cases, also
OH-DOAS data were available which have a generally better accuracy than LIF. Therefore, the DOAS data were used for the budget analysis (Fig. 10 - 12). Note that LIF and DOAS showed good agreement in these experiments (Fig. S2 and S3).

The rate constant for OH + $CH_4$ has an uncertainty factor of f=1.1 according to NASA/JPL 2020 and f=1.15 according to IUPAC web (4 June 2013). Thus, 10-15% difference in the radical budgets would be explainable by a too large rate constant.

This partly unresolved discrepancy shows the limitation of the analysis of radical production and destruction rates and
needs to be further investigated. The analysis of the experiment with the well-defined chemistry of methane indicates that the maximum accuracy of calculated differences between production and destruction rates for single radical species is 20 % in the chamber experiments.





## 6 Results and discussion of the experiments with myrcene

### 6.1 Reaction rate constant of the OH reaction with myrcene

The rate constant of the reaction of myrcene with OH is determined from the measured temporal decay of myrcene and the measured OH concentration. The time series of measured myrcene concentrations are compared to calculations using a chemical box model that only includes chemical loss reactions with OH and $O_3$ and dilution. The model is constrained to measured values of temperature, pressure, the dilution rate constant, OH and $O_3$ concentrations. The initial myrcene concentration is set to the injected amount of myrcene. The reaction rate constant for myrcene with $O_3$ is taken from the work by **?** who mea-

sured a value of $2.21 \times 10^{-15} \exp(-(520 \pm 109)K/T) \, \mathrm{cm^3 s^{-1}}$ using relative rate technique. Atkinson et al. (1986) suggested a nearly 20 % faster reaction rate constant of $4.7 \times 10^{-16} \, \mathrm{cm^3 s^{-1}}$ ($T = 298 \, \mathrm{K}$) that is still consistent with the value by **?** within the experimental uncertainties. The chemical lifetime of myrcene with respect to the ozonolysis reaction was approximately 44 min in the low $NO_x$ experiments with $O_3$ concentrations of 40 ppbv but was up to 3 hours in the experiments with medium $NO_x$ and $O_3$ concentrations of less than 10 ppbv. In comparison, the chemical lifetime of myrcene with respect to the reaction

with OH was within the range of 15 min at OH concentrations of approximately $5 \times 10^6 \, \mathrm{cm^{-3}}$ observed in all experiments. Hence, the contribution of ozonolysis to the total chemical loss of myrcene in the experiments was 10 % to 26 %.

The rate constant of the reaction of myrcene with OH is optimized, such that the difference between measured and modelled myrcene concentration time series is minimized. For this optimization the OH concentration observed from DOAS is used. This procedure is applied to all three experiments resulting in a rate constant of $(2.3 \pm 0.3) \times 10^{-10} \, \mathrm{cm^3 s^{-1}}$ ($T = 298 \, \mathrm{K}$, Table

4, Fig. S6 Supplement). The error results mainly from the variability of values determined in the different experiments.

The rate constant determined in this study is in good agreement with those reported in the literature (Table 4). Relative rate technique was used in experiments by Atkinson et al. (1986) and Grimsrud et al. (1975). The value reported by Hites and Turner (2009) $(3.4 \times 10^{-10} \mathrm{cm^3 s^{-1}})$ is higher than those of this work and reported by Atkinson et al. (1986), but has a high uncertainty of $\pm 35\%$ that is explained by experimental difficulties in the handling of myrcene. The rate constant calculated

from structure-activity relationship (SAR) by Peeters et al. (2007) is nearly 20 % lower than the experimentally derived reaction rate constants, but this difference is within the accuracy of SAR predictions.

### 6.2 Product yields of the reactions of myrcene hydroxy peroxy radicals with NO

When OH reacts with myrcene, about half (48 %) of the OH adds to the $-\mathrm{CH}{=}\mathrm{C(CH_3)_2}$ moiety and the other half (48 %) to the isoprenyl part (Peeters et al., 2007). In the first case, 4-vinyl-4-pentenal and acetone are formed from the reaction of $\mathrm{MyO_2}$

with NO (Section 2). In the second case, 2-methylidene-6-methyl-5-heptenal or 1-vinyl-5-methyl-4-hexenone are produced together with formaldehyde. Thus, the yields of acetone and formaldehyde are indicators for the yields of the OH addition to myrcene.

Acetone and formaldehyde yields are calculated from measured time series of product species and myrcene in the experiments on 16 and 22 August 2012, when NO mixing ratios were 200 pptv, so that >90 % of $RO_2$ reacted with NO (Fig. 5).

Only measurements after the first myrcene injection are used here to avoid that secondary chemistry impacts the yield deter-





mination. The two other experiments are not considered for three reasons. The large amount of ozone (up to 50 ppbv) would require considerable correction for myrcene ozonolysis. Second, due to the lower NO concentration, reactions of $RO_2$ with $HO_2$ would be competitive with the reaction of $RO_2$ with NO. Third, the smaller amount of injected myrcene produced less oxidation products.

Corrections are applied to measured myrcene, formaldehyde and acetone time series, in order to relate myrcene that reacted with OH and product species that are chemically formed from this reaction following the procedure described in previous work (Galloway et al., 2011; Kaminski et al., 2017). Product concentrations are corrected for losses due to dilution, photolysis and reaction with OH. In addition, formation of formaldehyde and acetone from chamber sources need to be subtracted from the measured concentrations. Myrcene concentrations are also corrected for dilution and the fraction of myrcene that reacted

with ozone. After corrections have been applied, the relationships between consumed myrcene and product concentrations are linear, if species are mainly formed as first-generation products in the reaction of myrcene and OH. The slopes give acetone and formaldehyde yields of $0.45 \pm 0.08$ and $0.35 \pm 0.08$, respectively (Fig. 8). The error is caused by the uncertainty in the source strengths of chamber sources for acetone and formaldehyde and the accuracy of measurements.

The acetone yield is in excellent agreement with SAR predictions that 48 % of the OH adds to the double bond of the

$-CH{=}C(CH_3)_2$ moiety (Peeters et al., 2007). The formaldehyde yield of $0.35 \pm 0.08$ is lower than the SAR prediction for the OH attack to the isoprenyl part (0.48). One possible reason for the smaller than expected formaldehyde yield could be isomerization reactions of the myrcene peroxy radicals shown (Fig. 3), which may not lead to the production of HCHO. The estimated bulk isomerization rates of $0.097\,s^{-1}$ and $0.21\,s^{-1}$ (Section 2) are sufficiently fast to compete with the loss rate of $0.04\,s^{-1}$ for $MyO_2$ with 200 pptv of NO, so that the reduced formaldehyde yield could be fully explained by isomerization

reaction pathways. A different explanation for the lower formaldehyde yield compared to acetone is that the nitrate yield is larger for the $MyO_2$ that would otherwise end up as formaldehyde. In fact, the total product yields is closed to unity (acetone $(0.45 \pm 0.08)$ + formaldehyde $(0.35 \pm 0.08)$ + nitrate $(0.13 \pm 0.03) = 0.93 \pm 0.12$).

Acetone and formaldehyde yields agree with published results by Reissell et al. (2002) and Orlando et al. (2000) (Table 5). Lee et al. (2006) reported a smaller acetone yield of $0.22 \pm 0.02$ and a significantly higher HCHO yield of $0.74 \pm 0.08$.

However, their HCHO yield carries a large uncertainty because concentrations were outside the range for which the instrument was calibrated.

The oxygenated organic compound 4-vinyl-4-pentenal has been detected in previous studies from the reaction of myrcene with OH (Reissell et al., 2002; Lee et al., 2006). Fragmentation in the PTR-MS and further oxidation of 4-vinyl-4-pentenal complicated the unambiguous yield determination. Therefore, Lee et al. (2006) reported a high yield of 0.4, but also state a

lower limit of 0.09. Acetone is the co-product of 4-vinyl-4-pentenal (Fig. 3). The high limit yield of 0.4 for 4-vinyl-4-pentenal by Lee et al. (2006) is therefore consistent with the acetone yield determined in this work. The yield of 4-vinyl-4-pentenal was also measured by Reissell et al. (2002), but a lower yield of $0.19 \pm 0.04$ was found. This low yield is apparently inconsistent with the yield of $0.45 \pm 0.06$ for the co-product acetone determined in the same experiments (Table 5). The authors suggest that there could be re-arrangement of the hydroxyalkoxy radical that compete with the decomposition to 4-vinyl-4-pentenal,

but may still lead to acetone production, so that the yield for acetone could become higher than that of 4-vinyl-4-pentenal.



The yield of organic nitrate from reactions of $MyO_2$ with NO is determined from the analysis of reactive nitrogen oxides in the chamber as described in Section 4. This results in an organic nitrate yield of $0.13 \pm 0.03$ (Fig. 9). This value is consistent with the yield of $0.10 \pm 0.03$ reported by Lee et al. (2006), who directly measured organic nitrates by mass spectrometry. Values are lower than the yield expected from SAR described in Jenkin et al. (2019), which predicts a yield of 0.19 that would also

apply for other monoterpenes, but may still be within the uncertainty of the SAR predictions. Like for the formaldehyde yield, the smaller than predicted organic nitrate yield may also be partly due to competing $MyO_2$ isomerization reactions.

### 6.3   Primary radical production and termination

The primary radical production is due to photolysis (Reactions R1 to R3) and ozonolysis reactions (Reaction R4). Production rates by photolysis of $O_3$, HONO and HCHO are calculated using measured trace gas concentrations and photolysis frequen-

cies. The calculation of the production rate from myrcene ozonolysis requires the knowledge of both the reaction rate constant (Section 6.1) and the yield of OH and $RO_2$ radicals. The uncertainty of yields are high. Values range from 0.71 determined from high-level quantum chemistry calculations and kinetic calculations (Deng et al., 2018) to experimental values of 1.15 with an uncertainty of a factor of 1.5 (Atkinson et al., 1992). It is worth noting that additional radicals could be produced from ozonolysis reactions of oxidation products at later times of the experiment because first-generation organic products from the

reaction of myrcene with OH still contain $C-C$ double bonds that can react with ozone.

The radical termination reactions include reactions with nitrogen oxides (Reactions R12-R14) and radical self-reactions (Reactions R15, R16). Loss of radicals in the reaction with $NO_x$ and the $HO_2$ self-reaction contribute less than $1\,ppbv/h$ to the total loss rate each. Contributions from $RO_2$ self-reactions are expected to be negligible ($< 2\,\%$) because their reaction rate constants are typically much smaller than those of $RO_2 + HO_2$ reactions (Table 3). The reaction rate constant of the $MyO_2 + HO_2$

reaction is estimated from SAR by Jenkin et al. (2019) and therefore, the value could also have a high uncertainty.

In the experiment with medium NO concentrations (22 August 2012), the total radical production is low with values of less than $1.5\,ppbv/h$ and radical production and destruction are balanced (Fig. 10). No ozone is added, so that radical production from ozonolysis plays a minor role, and radical loss by radical recombination is suppressed due to the competition of peroxy radical reactions with NO. Therefore, the uncertainties in the radical yield of myrcene ozonolysis and the reaction rate constant

of the $MyO_2 + HO_2$ reaction do not impact the results.

In contrast, radical production and destruction rates in experiments performed at low NO concentrations (17 July 2013, 18 July 2013) that is achieved by suppressing NO in the reaction with $O_3$ are dominated by radical production from myrcene ozonolysis and radical destruction by $MyO_2 + HO_2$ reactions right after the injection of myrcene. Taking the upper limit of the radical yield from the ozonolysis reaction of 1.42 (0.71 for OH and 0.71 for $MyO_2$), the production is $4\,ppbv/h$. The radical

primary production could be smaller, if the unspeficied radical yield of 1.15 suggested by Atkinson et al. (1992) is applied. Applying the reaction rate constant of the $MyO_2 + HO_2$ derived by SAR (Section 2) results in a maximum radical loss rate of $13.5\,ppbv/h$ also right after the injection of myrcene when $MyO_2$ concentrations are highest. Because this high destruction rate cannot be balanced by radical production, either the reaction rate constant must be lower than SAR predictions or there are reaction pathways that do not lead to radical termination products (organic peroxide), but regenerate radicals. Due to the





high uncertainty of radical production from ozonolysis in these experiments, no firm conclusion can be drawn. The reaction
rate constant would need to be reduced between a factor of 0.4 and 0.7 (0.9 to $1.6 \times 10^{-11} \, \mathrm{cm^3 s^{-1}}$ ($T = 298 \, \mathrm{K}$, Table 3) or
the yield of of radical products would need to be in the range of 0.3 and 0.6 to match the range of radical production.

A reduced $MyO_2 + HO_2$ reaction rate constant is significantly different from the SAR predictions by Jenkin et al. (2019). The
SAR value is similar for all monoterpenes and well agrees with direct measurements for the reaction rate constant of $HO_2$ with

$RO_2$ derived from $\alpha$-pinene ($2.1 \times 10^{-11} \, \mathrm{cm^3 s^{-1}}$), $\gamma$-terpinene ($2.0 \times 10^{-11} \, \mathrm{cm^3 s^{-1}}$) and limonene ($2.1 \times 10^{-11} \, \mathrm{cm^3 s^{-1}}$)
reported by Boyd et al. (2003). A high yield of radical products from the $MyO_2 + HO_2$ reaction may not be expected from
analogies of isoprene and 2-methyl-2-butene (Liu et al., 2012; Paulot et al., 2009). For these reasons there is no clear conclusion,
how exactly rate constants and yields need to be adjusted to balance the ROx production and destruction rates.

### 6.4   ROx radical chain propagation reactions

The total production and loss rates of OH, $HO_2$ and $RO_2$ radicals do not exhibit much variability over the course of the
experiments with medium NO concentrations (Fig. 11, Fig. S7 Supplement). In the experiments with low NO, the turnover
rates of radical production and destruction reactions for OH, $HO_2$, and $RO_2$ exhibit peak values when myrcene is injected
(Fig. 12 and Fig. S8 Supplement). This distinct feature is related to the elevated radical production of up to $2 \, \mathrm{ppbv/h}$ from the
ozonolysis of myrcene and peak concentrations of peroxy radical concentrations.

Production and destruction rates of OH radicals are balanced within the accuracies of the calculation in all experiments. Rates
are similar in all experiments with values around $4 \, \mathrm{ppbv/h}$. Maximum $RO_2$ turnover rates are lower (2 to 3 ppbv/h) compared
to those of OH and $HO_2$, because up to 25 % of the OH radicals are directly converted to $HO_2$, for example in the reaction of
OH with CO or HCHO. Because myrcene is consumed within a few hours, the turnover rate of $RO_2$ is decreasing over time
after each myrcene injection. In experiments with medium NO concentrations, $HO_2$ radical production and destruction rates

are not balanced by 0.5 of $1.5 \, \mathrm{ppbv/h}$ in contrast to those of OH. Discrepancies, however, are only slightly larger than the 1-$\sigma$
uncertainty of the calculated values. The total uncertainty is dominated by the accuracy of $RO_2$ measurements because of the
low detection sensitivity of the instrument for $MyO_2$ and the uncertainty in the sensitivity of other $RO_2$ formed from oxidation
products that are dominant, once myrcene reacted away.

Even higher discrepancies of up to $2 \, \mathrm{ppbv/h}$ between $HO_2$ and $RO_2$ production and destruction rates are observed in the

experiments with low NO mixing ratios. Values are highest right after each myrcene injection with the $HO_2$ production rate
being lower than the destruction rate and the $RO_2$ destruction rate being higher than the production rate. Because differences
are highest when the chemical system is dominated by the oxidation of myrcene and therefore the presence of $MyO_2$, reactions
pathways of $MyO_2$ other than reactions with NO and $HO_2$ could be responsible for the imbalances.

As described in Section 2, $MyO_2$ isomerization reactions can become competitive at low NO mixing ratios like in these

experiments ($< 0.1 \, \mathrm{ppbv}$). Isomerization rates derived from SAR are applied (Section 2). Products of the 1,6 H-shift reactions
in myrcene that correspond to similar reactions in isoprene could produce radicals like in the case of isoprene (Fig. 3). Products
from the two additional isomerization reactions likely undergo fast ring-closure reactions on the di-methyl double bond and it
is uncertain, if OH or $HO_2$ are directly produced. As an estimate for the potential impact of $MyO_2$ isomerization reactions on





radical regeneration, production of one HOx radical from each isomerization reaction is assumed in the following. The results

of this sensitivity analysis are shown in Fig. 11 and 12.

The bulk $MyO_2$ loss due to isomerization for the isomers that can rapidly interconvert (Fig. 2) are competitive with the reaction of $MyO_2$ with NO at mixing ratios of 0.5 to 1 ppbv making these reaction very competitive for conditions of the experiments. As shown in Fig. 12, the discrepancy between $RO_2$ radical production and destruction rates would be even over-compensated, if isomerization rates determined by SAR are applied. Sensitivity model runs suggest that a bulk isomerization

reaction rate of $0.05\,s^{-1}$ would be sufficient to balance the $RO_2$ production rate. This value is a factor 2 to 4 lower than bulk reaction rates calculated by SAR, but are still within the uncertainty of calculations.

Although this analysis shows the potential for high contributions of $MyO_2$ isomerization to the total loss rate of $RO_2$ for conditions that are typical for forested areas, the uncertainties are high concerning the isomerization rate constants and the yield of $HO_x$ radicals. In addition, discrepancies of radical production and destruction rates within the range of 20 % of the

total rate for single radical species can be observed even for well-known chemical systems as methane as shown in Section 5.2. This is likely due to uncertainties in the radical concentration measurements. Unaccounted systematic errors in the analysis may therefore also explain a significant fraction of the differences in radical production and destruction rates in the experiments with myrcene.

## 7   Summary and conclusions

The photooxidation of the monoterpene myrcene was investigated at atmospheric conditions in the simulation chamber SAPHIR at two different levels of NO. The chemical mechanism of the oxidation of myrcene by OH has not been investigated in detail so far. Based on the structural similarity with isoprene ($CH_2=CH-C(=CH_2)CH_2-$ moiety) and with 2-methyl-2-butene ($-CH_2-CH=C(CH_3)_2$ moiety) and structure-activity relationship (Peeters et al., 2007; Jenkin et al., 2019; Vereecken and Nozière, 2020) a chemical mechanism for the first oxidation step is proposed that includes rapid $RO_2$ interconversion by re-

versible oxygen addition and H-shift reactions of $RO_2$ like in isoprene. In addition to $RO_2$ isomerization reactions similar to that of isoprene derived $RO_2$, additional H-shift reactions are suggested by SAR, so that the overall impact of isomerization reactions in myrcene can be high, although only approximately half of the attack of OH to myrcene is on the isoprenyl part. Assuming that the interconversion of $RO_2$ is similar as in isoprene, bulk isomerization rate constants of $0.21\,s^{-1}$ and $0.097\,s^{-1}$ ($T = 298\,K$) for the 3 isomers resulting from the 3'-OH and 1-OH addition, respectively, are calculated using rate constants

derived by SAR.

Experiments in the chamber allowed to determine the reaction rate constant of the reaction of myrcene with OH resulting in a value of $(2.3\pm0.3)\times10^{-10}\,cm^3s^{-1}$, which is in good agreement with values reported in the literature within the uncertainties (Table 4). Product yields of acetone and formaldehyde calculated from measured time series of these species were found to be $0.45\pm0.08$ and $0.35\pm0.08$, respectively. Although these values well agree with studies by Reissell et al. (2002); Orlando

et al. (2000) within the uncertainties, the formaldehyde yield is lower than expected for conditions of the experiment with NO mixing ratios of 200 pptv, if there are no $RO_2$ isomerization reactions. The lower yield would be consistent with competing





$RO_2$ isomerization reactions that may not produce formaldehyde. From the analysis of nitrogen oxides in the chamber, the yield of organic nitrates in the myrcene oxidation could be determined to be $0.13 \pm 0.03$ in agreement with measurements by Lee et al. (2006) ($0.10 \pm 0.03$). Both values are lower than typical values predicted by SAR (0.19, Jenkin et al. (2019) and
found for monoterpene species which often range between 0.15 and 0.25. This may impact the formation of secondary organic aerosol from the OH oxidation of myrcene compared to other monoterpene species. The applicability of the procedure for the determination of the organic nitrate yield was tested in photo-oxidation experiments with methane and $\alpha$-pinene, both of which resulted in organic nitrate yields that well agree with expected values. This demonstrates that the method gives reliable results and can be applied in chamber experiments.

Radical production and destruction rates can be calculated using measured radical and trace gas concentrations. Total radical ($RO_x$) production and destruction rates are balanced in the experiments in this work, if the rate of the radical loss terminating the radical reaction chain due to recombination reactions of $HO_2$ and $RO_2$ was only 40 % of the loss expected from the reaction rate constant predicted by SAR or if this reaction would regenerate radicals instead of producing radical termination products (hydroperoxides). However, this conclusion has a high uncertainty of approximately a factor of 2 due to the uncertainty of
radical production from myrcene ozonolysis.

Imbalances between radical production and destruction rates are observed for $RO_2$ radicals in the experiments with low NO concentrations, when $MyO_2$ isomerization reaction can become competitive with bi-molecular reactions. Bulk reaction rates constants around $0.05\,\mathrm{s^{-1}}$ for MyO2 isomers which can quickly interconvert and which partly can isomerize, are a factor of 2 to 4 lower than reaction rates constants calculated by SAR, but values are consistent regarding the high uncertainty of the
determination from experiments and the uncertainty of SAR calculations.

The analysis of radical concentrations measurements in field campaigns, where monoterpene emissions dominated the mix of organic compounds such as in Finland (Hens et al., 2014) and in the Rocky Mountains (Kim et al., 2013) showed that current chemical models cannot explain measured OH and $HO_2$ concentrations. The analysis of chamber experiments revealed that the OH oxidation of the most abundant monoterpene species $\alpha$-pinene and $\beta$-pinene are not well understood (Kaminski
et al., 2017; Rolletter et al., 2019). Experiments here show that also the reaction of OH with myrcene is complex due to the rapid $RO_2$ inter-conversion and H-shift $RO_2$ reactions. Therefore, short-comings in the description of the photo-oxidation of myrcene could contribute to explaining model-measurement discrepancies found in field campaigns.

*Data availability.* Data are available from the Eurochamp database (16 August 2012: https://doi.org/10.25326/5XRG-Y765, Fuchs et al. (2021a), 22 August 2012: https://doi.org/10.25326/9RAV-5450, Fuchs et al. (2021b), 17 July 2013: https://doi.org/10.25326/GP2K-R926,
Fuchs et al. (2021c), 18 July 2013: https://doi.org/10.25326/JJ16-1S54, Fuchs et al. (2021d), 03 September 2019: https://doi.org/10.25326/PBBV-WF18, Fuchs et al. (2021e), 29 May 2020: https://doi.org/10.25326/MG6T-TW58, Fuchs et al. (2021f))





*Author contributions.* Z.T., L.H. and H.F. wrote the manuscript. M.K., R.W. and H.F. designed and led the experiments in the chamber. I.A. (organic compounds), B.B. (radiation), H.-P. D. (radicals), X.L. (HONO), S.N. (OH reactivity), F.R. (nitrogen oxides, ozone), R.T. (organic compunds), A.N. (radicals), C.C. (radicals) were responsible for measurements used in this work. All co-authors commented and discussed the manuscript and contributed thereby to the writing of the manuscript.


*Competing interests.* The authors declare that they have no conflict of interest.

*Acknowledgements.* This work was supported by the EU FP-7 program EUROCHAMP-2 (grant agreement no. 228335). This project has received funding from the European Research Council (ERC) under the European Union's Horizon 2020 research and innovation program (grant agreement no. 681529). S. Nehr and B. Bohn thank the Deutsche Forschungsgemeinschaft for funding (grant no. 1580/3-1). The authors thank Luc Vereecken for the discussion of the chemical mechanism.






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





**Table 1.** Experimental conditions for the oxidation experiments. Concentrations are given for the conditions in the SAPHIR chamber at the time of the first VOC injection.

| date | type of VOC | VOC / ppbv | NO / ppbv | O$_3$ / ppbv | $T$ / K |
|---|---|---|---|---|---|
| 16 August 2012 | myrcene | 2.3 | 0.2 | $< 20$ | 305 |
| 22 August 2012 | myrcene | 2.3 | 0.2 | $< 15$ | 310 |
| 17 July 2013 | myrcene | 0.8 | 0.06 | 40 | 305 |
| 18 July 2013 | myrcene | 0.8 | 0.09 | 40 | 308 |
| 29 May 2020 | methane | 140000 | 0.23 | 60 - 120 | 303 |
| 03 September 2019 | $\alpha$-pinene | 8.5 | 0.1 | 8 | 296 |





**Table 2.** Instrumentation for radical and trace gas detection in the experiments.

| measured quantity | measurement technique | time resolution | precision $(1\,\sigma)$ | accuracy $(1\,\sigma)$ |
|---|---|---|---|---|
| OH | differential optical absorption spectroscopy | 205 s | $0.6 \times 10^6$ cm$^{-3}$ | 6.5 % |
| OH | laser-induced fluoresence (LIF) | 47 s | $0.4 \times 10^6$ cm$^{-3}$ | 13 % |
| HO$_2$, RO$_2$ | laser-induced fluoresence (LIF) | 47 s | $1.5 \times 10^7$ cm$^{-3}$ | 16 % |
| MyO$_2$ | | | | 50 % |
| $k_{\mathrm{OH}}$ | laser photolysis + LIF | 180 s | 0.3 s$^{-1}$ | 10 %, $\pm 0.7$ s$^{-1}$ |
| myrcene, $\alpha$-pinene | proton-transfer-reaction mass-spectrometry | 40 s | 15 pptv | 7 % |
| myrcene, acetone | gas-chromatography | 30 min | 100 pptv | 6 % |
| acetaldeyhde | | | | |
| CH$_4$ | cavity ring-down spectroscopy | 60 s | 1 ppbv | 3 ppbv |
| CO | cavity ring-down spectroscopy | 60 s | 25 ppbv | 1 ppbv |
| NO | chemiluminescence | 180 s | 4 pptv | 5 % |
| NO$_2$ | photolytic converter + chemiluminescence | 180 s | 2 pptv | 5 % |
| HONO | long-path absorption photometry | 300 s | 7 pptv | 20 % |
| O$_3$ | UV-absorption | 10 s | 1 ppbv | 5 % |
| HCHO | differential optical absorption spectroscopy | 100 s | 20 % | 10 % |
| HCHO | Hantzsch method (Aerolaser) | 90 s | 100 pptv | 10 % |
| photolysis freq. | actinic flux spectroradiometry | 60 s | 10 % | 10 % |



**Table 3.** Chemical reactions considered in the radical budget analysis of OH, $HO_2$ and $RO_2$. Some reactions are only applicable in the experiment with either methane, $\alpha$-pinene or myrcene, in which specific $RO_2$ radicals, $CH_3O_2$, $MyO_2$ or $APO_2$ are formed. $RO_2+RO_2$ reactions are not listed, because they do not significantly contribute to the loss of radicals for conditions of experiments here.

| | Reaction | $k$(T=298 K, P=1 atm ) | uncertainty[a] | reference |
|---|---|---|---|---|
| **radical initiation reactions** | | | | |
| R1 | $HONO+h\nu$ (< 400 nm) → OH+NO | $j_{HONO}$ | 22 % | measured |
| R2 | $O_3+h\nu$ (< 340 nm) → $O(^1D)+O_2$ | $j_{O(^1D)}$ | 11 % | measured |
| | $O(^1D)+H_2O$ → OH+OH | $2.1 \times 10^{-10}$ cm$^3$s$^{-1}$ | | IUPAC (2020) |
| | $O(^1D)+M$ → $O(^3P)+M$ | $3.3 \times 10^{-11}$ cm$^3$s$^{-1}$ | | IUPAC (2020) |
| R3 | $HCHO+h\nu$ (< 335 nm)$+2O_2$ → $2HO_2+CO$ | $j_{HCHO}$ | 11 % | measured |
| R4 | $O_3+myrc$ → 0.71 OH+0.71 $RO_2$ | $3.9 \times 10^{-16}$ cm$^3$s$^{-1}$ | 22 % | Deng et al. (2018); **?** |
| **radical propagation reactions** | | | | |
| R5 | $HO_2+NO$ → OH+ $NO_2$ | $8.5 \times 10^{-12}$ cm$^3$s$^{-1}$ | 27 % | IUPAC (2020) |
| R6 | $HO_2+O_3$ → OH+ 2 $O_2$ | $2.0 \times 10^{-15}$ cm$^3$s$^{-1}$ | 20 % | IUPAC (2020) |
| R7 | $OH+VOC+O_2$ → $RO_2$ | | | |
| | $RO_2 = CH_3O_2$ | $6.4 \times 10^{-15}$ cm$^3$s$^{-1}$ | 17 % | IUPAC (2020) |
| | $RO_2 = MyO_2$ | $2.3 \times 10^{-10}$ cm$^3$s$^{-1}$ | 25 % | this work |
| R8 | $OH+HCHO+O_2$ → $HO_2+CO$ $+H_2O$ | $8.4 \times 10^{-12}$ cm$^3$s$^{-1}$ | 25 % | IUPAC (2020) |
| R9 | $OH+CO+O_2$ → $HO_2 + CO_2$ | $2.3 \times 10^{-13}$ cm$^3$s$^{-1}$ | 17 % | IUPAC (2020) |
| R10 | $OH+O_3$ → $HO_2+O_2$ | $7.3 \times 10^{-14}$ cm$^3$s$^{-1}$ | 17 % | IUPAC (2020) |
| R11 | $RO_2+NO$ → $HO_2+NO_2+R'CHO$ [c] | | | |
| | $RO_2 = CH_3O_2$ | $7.7 \times 10^{-12}$ cm$^3$s$^{-1}$ | 27 % | IUPAC (2020) |
| | $RO_2 = MyO_2$ | $0.87 \times 9.1 \times 10^{-12}$ cm$^3$s$^{-1}$ | 55 % | IUPAC (2020)[b] |
| **radical termination reactions** | | | | |
| R12 | $OH+NO_2$ → $HNO_3$ | $1.2 \times 10^{-11}$ cm$^3$s$^{-1}$ | 28 % | IUPAC (2020) |
| R13 | $OH+NO$ → HONO | $9.8 \times 10^{-12}$ cm$^3$s$^{-1}$ | 28 % | IUPAC (2020) |
| R14 | $RO_2+NO$ → $RONO_2$ | | | |
| | $RO_2 = CH_3O_2$ | $\approx 0$ | - | IUPAC (2020) |
| | $RO_2 = APO_2$ | $0.18 \times 9.1 \times 10^{-12}$ cm$^3$s$^{-1}$ | 27 % | IUPAC (2020)[b] |
| | $RO_2 = MyO_2$ | $0.13 \times 9.1 \times 10^{-12}$ cm$^3$s$^{-1}$ | 55 % | IUPAC (2020)[b] |
| R15 | $RO_{2,myrc}+HO_2$ → ROOH $+O_2$ | $9.1 \times 10^{-12}$ cm$^3$s$^{-1}$ | 25 % | this work |
| R16 | $HO_2+HO_2 + H_2O$ → $H_2O_2 +O_2 + H_2O$ | $4.0 \times 10^{-13}$ cm$^3$s$^{-1}$ [d] | 25 % | IUPAC (2020) |

[a] Uncertainty measurements (Table 2) and kinetic rate constants (10 % for reactions of inorganic and methyl peroxy radicals and 15 % for other)

[b] Reaction rate constant: IUPAC (2020); the branching ratio of 0.87 is taken from this work

[c] The dominant reaction of alkoxy radical is to form $HO_2$ which is not rate limiting for $RO_2$ to $HO_2$ conversion

[d] Effective reaction rate constant for 1 % $H_2O$ mixing ratio



**Table 4.** Rate constants of the myrcene + OH reaction at 298 K.

| $k\,\mathrm{cm^3 s^{-1}}$ | method | reference |
|---|---|---|
| $(2.1 \pm 0.2) \times 10^{-10}$ | relative rate technique | Atkinson et al. (1986) |
| $(3.4^{+1.5}_{-1.0}) \times 10^{-10}$ | relative rate technique | Hites and Turner (2009) |
| $1.8 \times 10^{-10}$ | structure-activity relationship | Peeters et al. (2007) |
| $(2.3 \pm 0.3) \times 10^{-10}$ | direct measurement | this work |



**Table 5.** Products yields from the reaction of myrcene with OH.

| product | yield | temperature / K | reference |
|---|---|---|---|
| formaldehyde | $0.30 \pm 0.06$ | n.a. | Orlando et al. (2000) |
| | $0.74 \pm 0.08$ | 294 | Lee et al. (2006) |
| | $0.35 \pm 0.08$ | 293-330 | this work |
| acetone | $0.36 \pm 0.05$ | n.a. | Orlando et al. (2000) |
| | $0.45 \pm 0.06$ | $296 \pm 2$ | Reissell et al. (2002) |
| | $0.22 \pm 0.02$ | 294 | Lee et al. (2006) |
| | $0.45 \pm 0.08$ | 293-330 | this work |
| organic nitrate | $0.10 \pm 0.03$ | 294 | Lee et al. (2006) |
| | $0.13 \pm 0.03$ | 293-330 | this work |



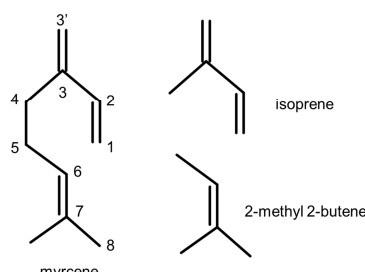

**Figure 1.** Chemical structures of myrcene, isoprene and 2-methyl 2-butene.



**Figure 2.** Peroxy radicals formed in the addition of OH to the double bonds in myrcence. $MyO_2$ formed from the OH addition to the isoprenyl part of myrcene leads to the formation of $MyO_2$ species that are expected to rapidly interconvert by oxygen abstraction and elimination like in isoprene (Peeters et al., 2014). Assuming reaction rate constants for oxygen abstraction and elimination like in isoprene, $3'-OH-3-OO$ and $1-OH-2-OO$ are expected to have the highest yields among the $MyO_2$ species that can interconvert. $MyO_2$ yields are predicted using SAR by Peeters et al. (2007).





**Figure 3.** Reactions of the four most abundant myrcene hydroxy peroxy radicals with NO forming $HO_2$ and carbonyl compounds.





**Figure 4.** Fast H-shift reactions of myrcene peroxy radicals. Rate constants apply to 298 K. They are adopted for the $Z-3'-OH-1-OO$ and $Z-1-OH-3'-OO$ isomers from the corresponding 1,6-H-shift reactions of isoprene peroxy radicals (Peeters et al., 2014). Similar subsequent chemistry like for isoprene leads to the formation of peroxy radicals, some of which rapidly decompose. The isomerization reaction rate constants for the $E-3'-OH-1-OO$ and $E-1-OH-3'-OO$ peroxy radicals are calculated using SAR by Vereecken and Nozière (2020). Subsequent chemistry of products from these reactions likely undergo fast ring-closure reactions on the dimethyl double bond (not shown here).



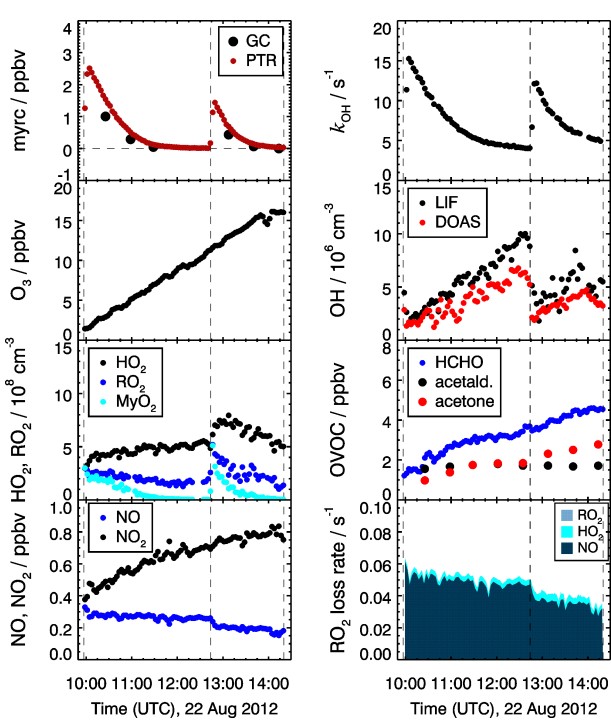

**Figure 5.** Trace gas and radical concentrations in the chamber experiment investigating the OH oxidation of myrcene (myrc) at NO mixing ratios between 200 and 300 pptv on 22 August 2012. Peroxy radicals from the reaction of OH with myrcene ($MyO_2$) and total organic peroxy radical ($RO_2 = MyO_2 + RO_{2,other}$) concentrations are calculated from observed radical concentrations as explained in Section 3.4. In the lowest right panel, the loss rates of total $RO_2$ with respect to their reaction with NO and radical recombination reactions are shown.



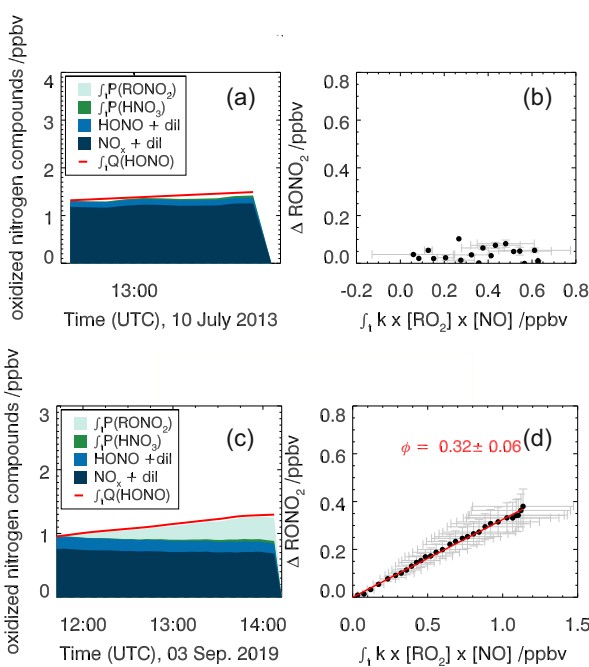

**Figure 6.** Time series of nitrogen oxide species in the experiments with methane (10 July 2013, (a)) and $\alpha$-pinene (03 September 2019, (c)). The red lines in panels (a) and (c) are time integrated HONO emissions calculated as described in the text. $NO_x$ and HONO concentrations were measured, the time integrated $NO_2$ loss by $HNO_3$ formation from the reaction of $NO_2$ with OH and the time integrated $NO_2$ loss by $RONO_2$ formation from the reaction of $RO_2$ with NO were calculated as described in the text. The alkyl nitrate yield ($\Phi_{RONO2}$) is determined from the regression analysis (red line, panel (d)) using Equation 8. Errors are derived from error propagation of measurements.

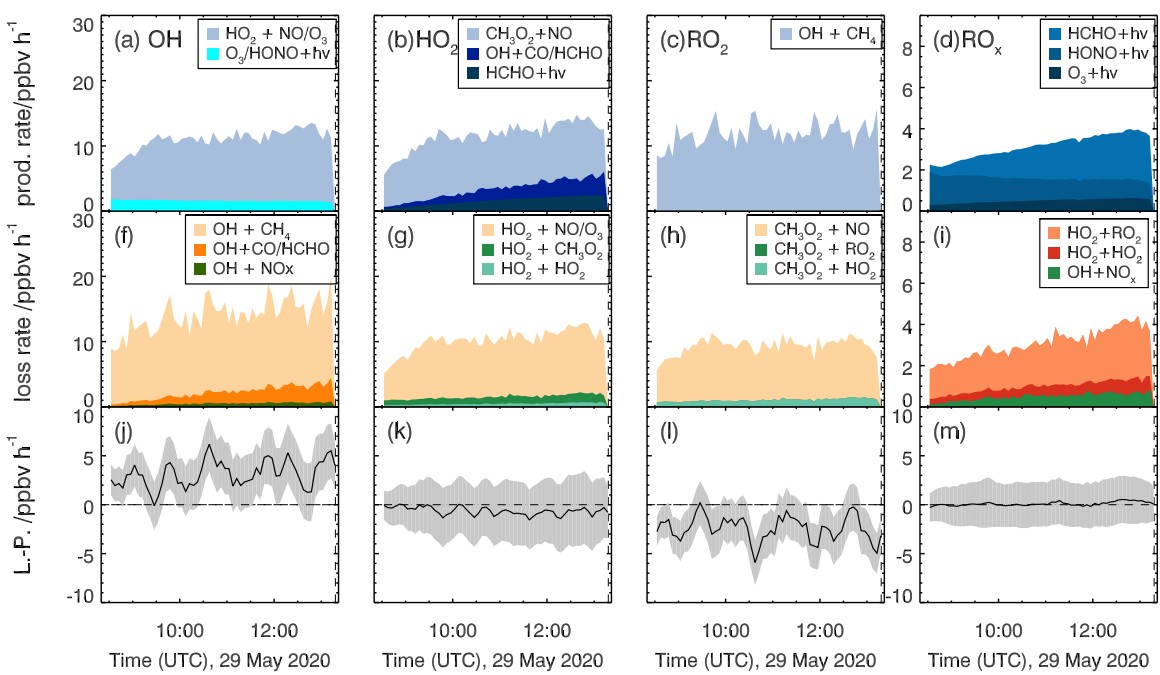

**Figure 7.** Calculated production and destruction rates of reactions involving radicals for the reference experiment with methane (29 May 2020). Production rates for OH, $HO_2$, $RO_2$ and total $RO_x$ are shown in panels (a), (b), (c) and (d), respectively and loss rates for OH, $HO_2$, $RO_2$ and total $RO_x$ are shown in panels (f), (g), (h) and (i). Lower panels (j) – (m) show the differences between loss and production rates $(L - P)$ with the associated accuracy (grey areas).

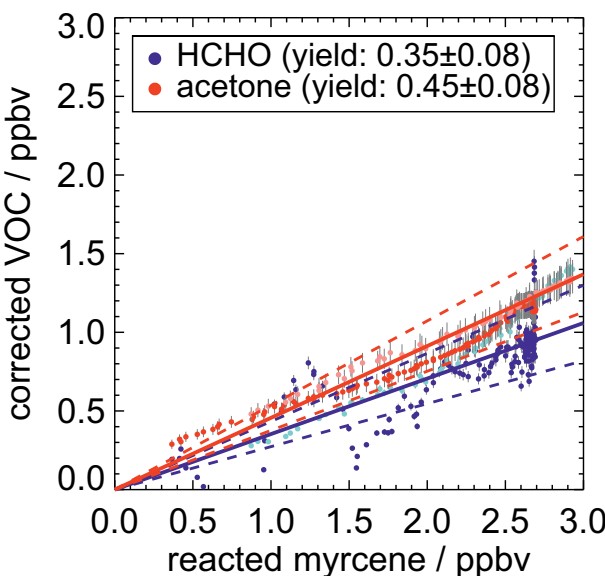

**Figure 8.** Corrected product concentrations versus the myrcene that reacted away with OH for two experiments with medium NO (light colors: 16 August 2012, dark colors: 22 August 2012). Corrections are applied to account for formation of product species not connected to the oxidation of myrcene (chamber sources) and loss processes (reaction with OH, photolysis). The slopes of linear fits (solid lines) give the product yields of HCHO and acetone from the myrcene reaction with OH. The uncertainties of the yield calculations are indicated by dashed lines.

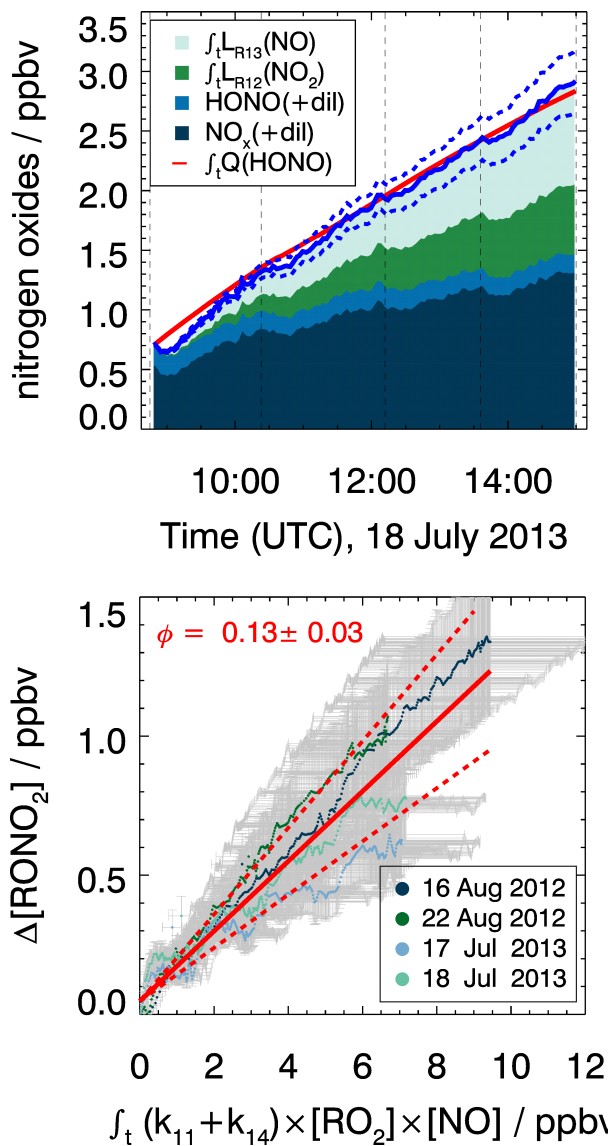

**Figure 9.** Determination of the organic nitrate yield from the production of nitrogen oxide species ($Q(\mathrm{HONO})$) in the chamber experiment. Upper panel: Example of cumulated reactive nitrogen oxide mixing ratios over the course of the experiment on 18 July 2013. The blue line denotes the total production of nitrogen oxides including organic nitrate applying a nitrate yield ($\Phi_{\mathrm{RONO2}}$) of 0.13. Dashed lines show the error of the total nitrogen oxide calculation. Lower panel: Scatter plot of integrated turnover rate of the reaction of $\mathrm{RO_2}$ with NO versus the unaccounted nitrogen oxide mixing ratios ($\Delta\mathrm{NOy}$, Equation 8). Results from all four experiments are included. The organic nitrate yield is determined by the slope of regression analysis (red line, $R^2 = 0.92$). Error bars denotes experimental uncertainty derived from the accuracy of the measurements (Table 2) and dashed red lines the resulting uncertainty in the slope of the regression analysis.

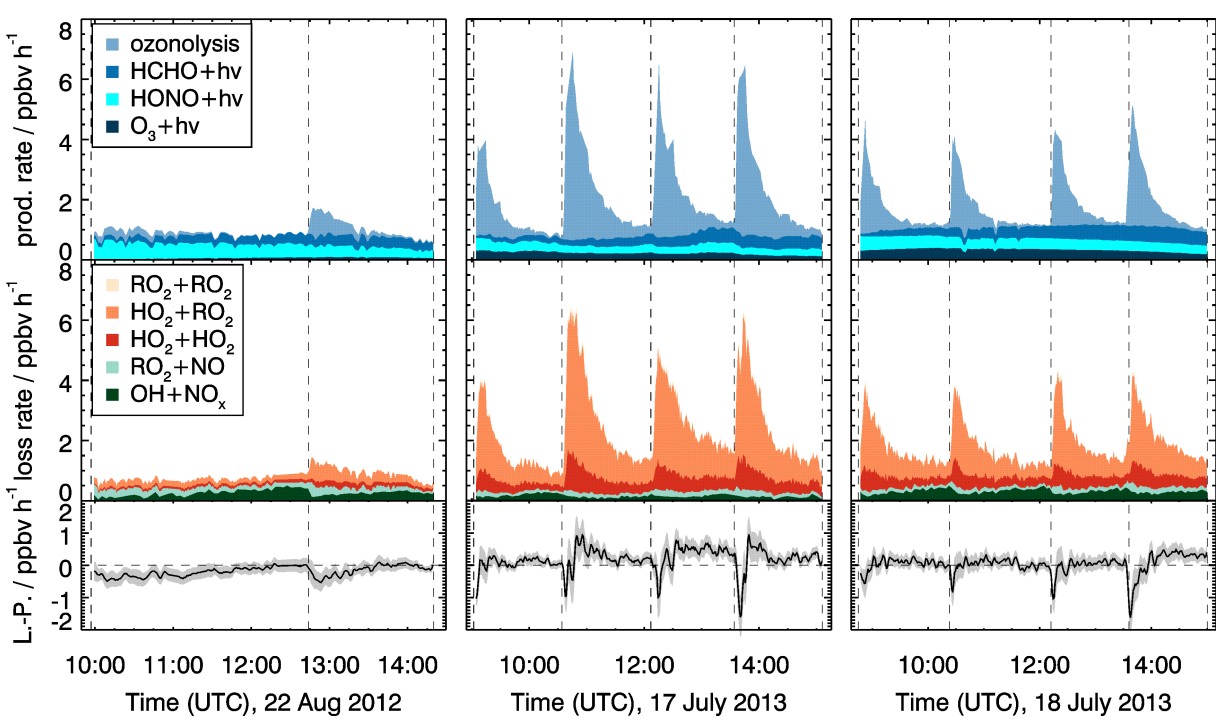

**Figure 10.** $RO_x$ primary production, $P$, (upper panels), termination $L$ (middle panels) rates and their difference (lower panels). Grey areas in the lower panels give the uncertainty of the difference between radical destruction and production ($L - P$). The rate constant of the reaction of $MyO_2$ with $HO_2$ was adjusted, to minimize the difference between radical production and destruction (see text for more explanation).

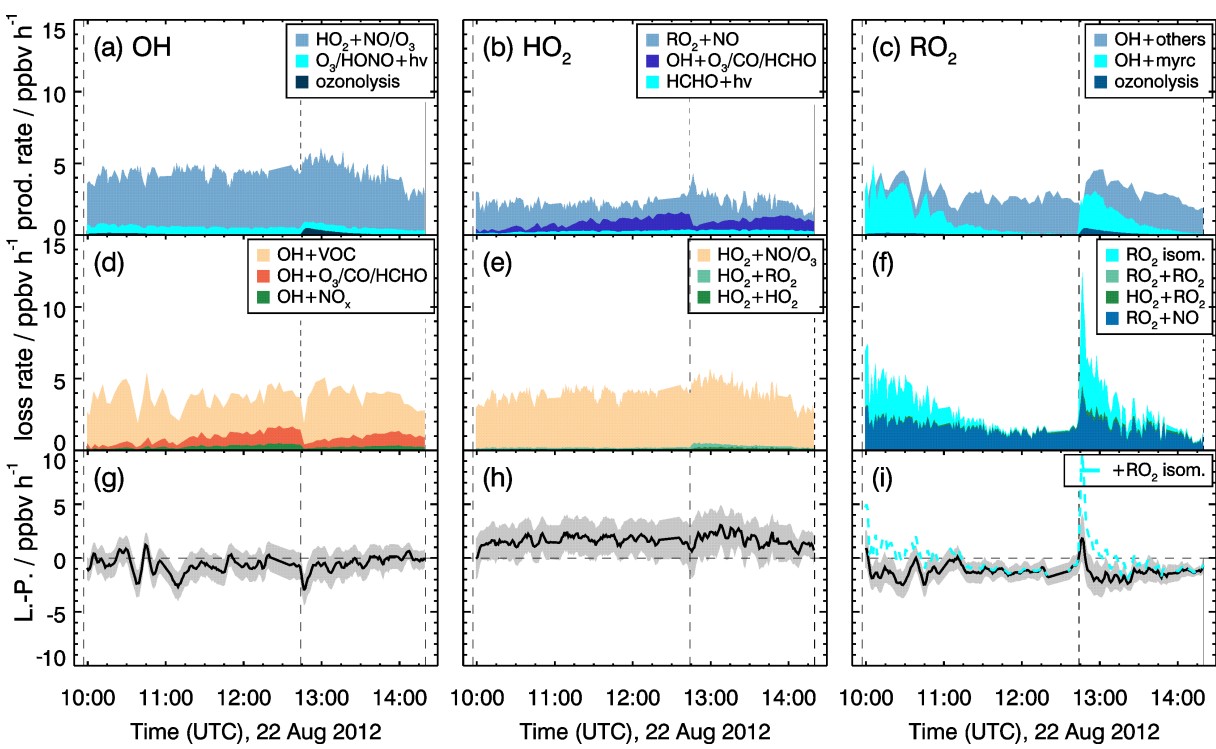

**Figure 11.** Rates of radical conversion reactions and imbalances between production and destruction rates ($L - P$) for the experiment with medium NO mixing ratios (0.15 to 0.30 ppbv) on 22 August 2012. Grey areas in the lower panels give the uncertainty of the calculation ($L - P$). In the lowest right panel, the black and cyan lines denote $RO_2$ loss rates without and with considering $MyO_2$ isomerization reactions.

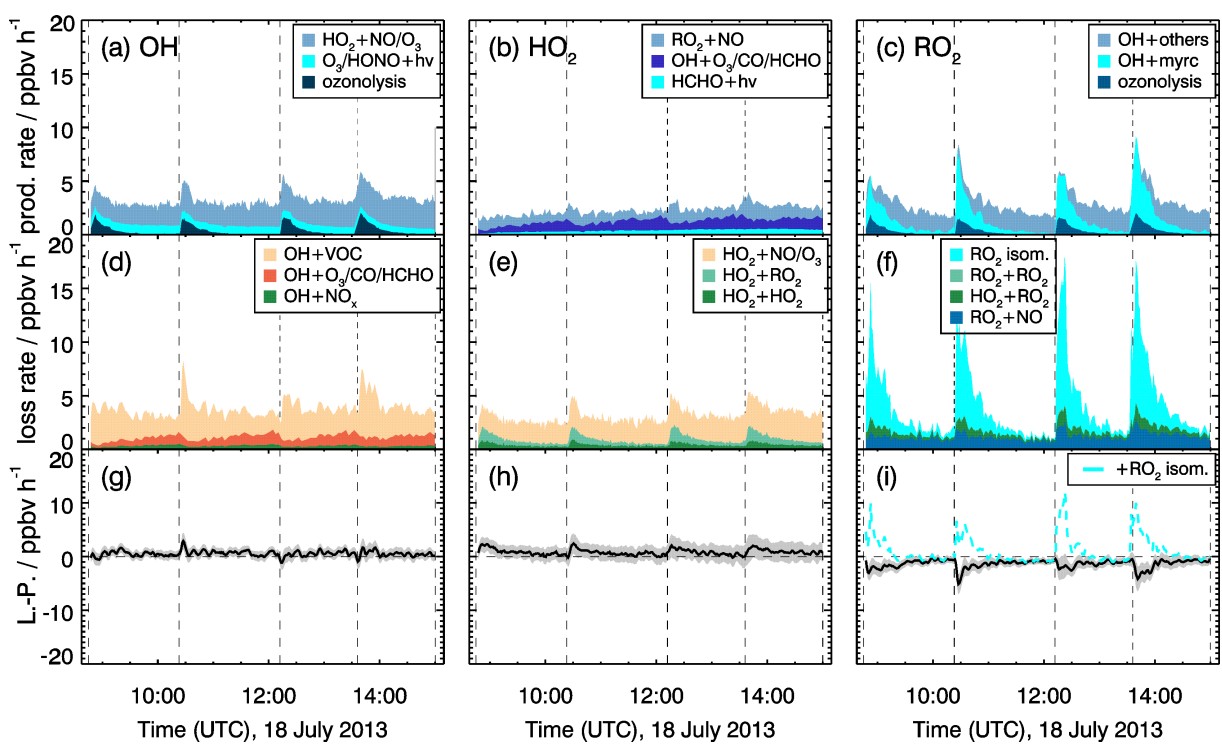

**Figure 12.** Rates of radical production and destruction reactions for OH, $HO_2$ and $RO_2$ for experiments at low NO mixing ratio (<0.11 ppbv) on 18 July 2013. Grey areas in the lower panels give the uncertainty of $L - P$. In the lowest right panel, the black and cyan lines denote $RO_2$ budget without and with considering $MyO_2$ isomerization reactions.