# Peer review of "Supporting Information to "Atmospheric photo-oxidation of myrcene: OH reaction rate constant, gas phase oxidation products and radical budgets""

_Atmospheric Chemistry and Physics, 2021_

## Referee Comment (RC1)

The authors use a highly instrumented outdoor atmospheric simulation chamber in order to investigate the mechanism of beta-myrcene photooxidation under a range of atmospherically relevant conditions. Beta-myrcene is an important biogenic $C_{10}$ terpene, containing 3 C-C double bonds, 2 of which are conjugated (c.f. isoprene), and therefore it's atmospheric degradation chemistry is complex, and hence not well understood, as highlighted in this study.

The authors use an impressive range of analytical techniques for measuring the decay of precursor compounds, oxygenated product formation, and, most importantly, a range of radical intermediate species in order to gain kinetic and mechanistic insights into myrcene's atmospheric chemistry. An early stage mechanism for myrcene + OH is built from literature SAR calculations and evaluated using a number of innovative kinetic analyses and a range of important product yields elucidated in the experiments.

The work described in this paper is well conceived and carried out, with the substantial uncertainties involved in the measurements well thought out and discussed. The results are presented are extremely valuable to the atmospheric chemistry community, giving important insights into complex monoterpene atmospheric chemistry and providing a framework with which to investigate other complex atmospheric chemistry mechanisms and to test and optimise the SARs used to construct atmospheric model mechanisms

I recommend this work for publication in ACP after a few minor comments/suggestions are addressed which should aid and improve the impact and clarity of this paper.

PG 3: Oxidation mechanism of myrcene: The SAR of Peeters 2007 has recently been used and updated in the protocol by Jenkin et al., 2018 on the "Estimation of rate coefficients and branching ratios for gas-phase reactions of OH with aliphatic organic compounds for use in automated mechanism construction". Have you looked at this to see if it gives different results to Peeters 2017? Never the less, the Jenkin et al., 2018 work should be citied here also.

Could epoxide formation in this system (c.f. IEPOX from isoprene + OH) also be an important pathway in the myrcene mechanism, with important implications on SOA formation?

PG 3: Spelling: "epxerimental" and "Peters"

PG 4, line 100: "Products likely undergo fast ring-closure reactions on the dimethyl double bond with rates on the order of 1 s−1 (pers. comm. Vereecken, 2021b)" – why is this chemistry not included in Fig. 4?

PG 4, line 111: Should "Fig 1" be "Fig 2"?

PG 5, line 129: "e.g." wrong place

PG 5, line 130: add the experiment RH values to Table 1

PG 5, line 131: define "medium" NO in the context of your experiments

PG 5, line 150: "Approx 50 ppbv O3 added..." - therefore ozonolyiss of myrcene will form products similar to the OH reaction, potentially interfering with the mechanistic results. This needs to be discussed further here, and you need to show evidence that interferences from ozonolysis is minimized in these experiments (which you discuss later).

PG 5, line 142 – 145: I am unsure how these concentrations discussed here map onto those given in Table 1, i.e. 2.3 ppbv for the first injection in the "medium" NO expts and 0.8 ppbv for the first injection in the lower NO experiments...?

PG 5, line 146 – 149: again, cross reference the concentration data with the data given in Table 1

PG 6, line 163: How do the different methods of measuring HCHO compare? ( and show evidence they are comparable in the experiments here, or that they are comparable from other SAPHIR experiments)

PG 7, line 197: VOC reactivity and RO2 speciation in myrcene experiments – give a brief explanation of how k'(OH) was measured here

PG 11, line 313: you need to provide information on where the ozonolysis OH and RO2 yields are derived from in Table 3. What about the ozonolysis yields of HO2?

PG 13, line 378: "reacivity"

PG 13, line 379: Why was the methane experiment, which is key to understanding the uncertainties of this analysis, not repeated with the DOAS OH measurements? (and ideally a different CH4 measurement?)

Could experiments on a simple, well known alkene + OH system (such as ethene or TME) also be useful here?

PG 13, line 386: Both JPL and IUPAC need to be referenced appropriately

PG 14, line 399 and 401: appropriately reference "?"

PG 14, line 415: Jenkin et al., 2018 gives an My + OH rate constant (298 K) of 1.88E-10 cm-3 s-1 (see earlier comment with respect to update to the Peeters 2007 SAR...)

PG 15, line 440: Derive the yields from Jenkin et al., 2018 as well. Could you not also use a simple model to show the impact of RO2 isomerisation reactions on the carbonyl yields?

Which instrument(s) was used to measure the HCHO yields in the different experiments here?. how do the different methods compare?

PG 17, line 499: "well agrees"

PG 18, line 534: "Sensitivity model runs"  Explain how these models were built and run here

Table 2:  "acetaldehyde" (spelling):  How and why was this measured here?

Table 3:  R4 - ?

Table 4:  Add the value derived from Jenkin et al., 2018 – 1.88E-10 cm3 s1

Figure 8:  This figure is a bit messy, and could be clearer. Separate into 2 Figures (i.e. Figure 8a and Figure 8b)

**Reference**

Jenkin, M. E., *et al.,* Estimation of rate coefficients and branching ratios for gas-phase reactions of OH with aliphatic organic compounds for use in automated mechanism construction., *Atmos. Chem. Phys*., **18**, 9297–9328, 2018 .https://doi.org/10.5194/acp-18-9297-2018